# CAN MODELS LEARN FROM ARBITRARY PAIRS?

## ABSTRACT

Representation learning traditionally follows a simple principle: pull semantically similar samples together and push dissimilar ones apart. This principle underlies most existing approaches, including supervised classification, self-supervised learning, and contrastive methods, and it has been central to their success. Yet it overlooks an important source of information: Even when classes appear unrelated, their samples often share latent visual attributes such as shapes, textures, or structural patterns. For example, cats, dogs and cattle have fur and four limbs etc. These overlooked commonalities raise a fundamental question: *can models learn from arbitrary pairs without explicit guidance?*

We show that the answer is yes. The primary challenge lies in learning from dissimilar samples while preserving the notion of semantic distance. We resolve this by proving that for any pair of classes, there exists a subspace where their shared features are discriminative to other classes. To uncover these subspaces we propose **SimLAP**, a **Sim**ple framework to **L**earn from **A**rbitrary **P**air. SimLAP uses a lightweight feature filter to adaptively activate shared attributes for any given pair. Through extensive experiments we show that models trained via SimLAP can indeed learn effectively from arbitrary pairs. Remarkably, models learned from arbitrary pairs are more transferable than those learned from traditional representation learning methods and exhibit greater resistance to representation collapse. Our findings suggest that arbitrary pairs, often dismissed as irrelevant, are in fact a rich, complementary and untapped source of supervision. By learning from them we move beyond rigid notions of similarity. Hopefully, SimLAP will open an additional pathway toward more general and robust representation learning.

## 1 INTRODUCTION

The basic tenet of most of the tasks in machine learning, such as classification, clustering, image segmentation, or self-supervised learning, is the notion of similarity or dissimilarity. They both serve to highlight unique characteristics of objects or concepts. While similarities emphasise commonalities, dissimilarities focus on distinctive characteristics. Both notions are essential for understanding and categorising the world around us.

Conventional machine learning models seek representation which aims to maximise the similarity of instances belonging to the same class, cluster or segment, while maximising dissimilarities of samples drawn from different categories. The success is typically measured using cross-entropy (LeCun et al., 1989; Khalid et al., 2023; Sohn, 2016), when the learning is assisted by labels (Khosla et al., 2020) or some other contrastive learning (CL) loss function, when labels are not available (Chen et al., 2020; He et al., 2020; Atito et al., 2021). During training, the models fit weight vectors to the respective groups of data so that samples from the same group are pulled together, while samples from different classes are pushed apart (Liu et al., 2016; Wu & Wang, 2020). This objective focuses on discrimination of one group against all the others, and implicitly promotes orthogonal representations (Ranasinghe et al., 2021). However, this one against all approach ignores the potential relationships between different groups, even though it is evident that classes may share common features. For example, cats and dogs are distinct classes but share mammalian traits, such as four limbs, fur, and tail. Even for distinct classes, snakes and ropes both follow elongated curves. The conventional one-against-all learning fails to consider these relationships.

Recently, a resurgence of work has been aware of the importance of common features across classes (Cabannes et al., 2023b; Sobal et al., 2024; Ridnik et al., 2021). These interesting methods apply

additional sources of information to build a better definition of class similarity. However, it is infeasible to provide this Aristotelian annotation for all class pairs because of their tremendous number in datasets, such as ImageNet-1K (Deng et al., 2009). Secondly, such auxiliary information, in the form of e.g. semantic hierarchy (Ridnik et al., 2021), has limitations in annotating the relationship between classes — birds and planes are similar in body shape but semantically distant from each other.

In view of these comments, we propose a completely new paradigm of learning refered to as SimLAP (a Simple framework for Learning from Arbitrary Pairs). Instead of maximising within group and minimising between group similarity, we aim to optimise the *between group similarity which is distinct for each pair of groups (classes)*. This will automatically identify features that are shared by pairs of classes. In the context of our proposition, the first fundamental question is how such learning can be formulated. If we take any pair of groups or classes, we still want them to be distinguishable, that is to be dissimilar, at the end of learning. Hence, any similarity between the pair can possibly exist only in a subspace. Thus our desire to learn similarity between any pair of distinct classes can be realised only by identifying the appropriate subspace for each pair. In conclusion, our learning problem has to be stated as one of searching for subspaces where the similarity of the corresponding class pairs is maximised.

The proposed formulation of learning is fundamentally different from the conventional one-against-all approach pursued to date. It is directed to provide much richer information about the unique groups of data in the training set. However, it raises a second fundamental question: *Can it learn to separate each individual group from all the other groups as well?* In other words, if we learn that a pair of groups, e.g. dogs and cats have shared features (four limbs), will we be able to separate the groups as well. We demonstrate experimentally in Sections 4 and 5 that this is a viable learning method, that achieves both objectives. It leads to one-against-all separability, as well as providing much richer information about the relationship of any pair of classes in our training set. This contrast with the supervised learning methods that do attempt to reflect class relationships, e.g., Supcon (Khosla et al., 2020), which fail to learn orthogonal representations when assembling distinct pairs as shown in Section 5 Figure 6.

We demonstrate our methodology of learning from arbitrary pair on IN1K. Its 1,000 classes comprise 500,500 (0.5M) pairs, with no explicit annotation regarding their pairwise similarity. To find the requisite class-pairwise subspaces, we introduce a simple module, *feature filter*, to create a subspace for each pair. For a chosen class pair, this adaptive filter learns a gate vector (values between 0 and 1) to identify the subspace in which their samples are discriminative to other class pairs. Our model SimLAP can learn visual representations and subspaces for class pairs simultaneously by gradient descent. We show that SimLAP achieved competitive performance with the models trained conventionally, using the one-against all approach based on separating similar samples from the rest.

The paper is organized as following: Section 3.1 demonstrates a hard example of extracting discriminative features in a subspace, which can be proved in Appendix A.1. Motivated by this idea of learning in subspaces, we propose SimLAP in Section 3.3. We build a simple and clean implementation to better highlight the contributions from arbitrary pairs in Section 4. Although our objective promotes compactness of distinct pairs in subspaces, SimLAP successfully exhibits class separability in global space, cf. Figure 5. Section 5 demonstrate the following experimental findings : Our gating mechanism is the key of robust learning from distinct pairs cf. Figure 6. SimLAP is driven by a learning dynamic of representation learning and subspace optimization, cf. Table 4. Furthermore, SimLAP exhibits amazing properties in preventing collapse cf. Figure 7. We improve SimLAP with a momentum encoder to demonstrate the advantages in Appendix C. Additionally, SimLAP is complementary to existing learning mechanisms and validate this by joint loss and mid-training, cf. Section 6. Finally, we argue that SimLAP is complementary to existing learning mechanisms and validate this by applying SimLAP in joint loss and mid-training, cf. Section 6. In summary, our contributions are as follows: ❶ We propose a novel paradigm of learning from pairs of samples drawn from an arbitrary pair of classes that groups them in their corresponding subspace. ❷ We show that the models learned from arbitrary pairs identify the features shared by each pair. ❸ Most importantly, a clean SimLAP learn to separate individual classes from all the other classes. The performance is not significant worse than supervised learning, indicating arbitrary pairs is as important as identical pairs. ❹ Interestingly, SimLAP is complementary to existing learning principles. Both combining with instance discrimination (MoCo) and masked image modelling (MAE) achieves performance improvement.

## 2 RELATED WORK

**Supervised Learning.** Existing visual representation learning (VRL) paradigms, whether supervised learning (SL) or self-supervised learning (SSL), follow the intuition that learning objectives should promote a predefined similarity structure (Cabannes et al., 2023a). The representations implicitly learn the notion of similarity by pulling semantically identical pairs closer while pushing semantically non-identical or distinct pairs away as a whole either directly or indirectly (Liu et al., 2016; Wang et al., 2016). Despite the prevailing success of these VRL paradigms, they lack a systematic mechanism to explicitly learn from semantically distinct pairs, regardless of the semantic distance between these distinct pairs. Recent work finds that semantically distinct but visually similar samples can help in representation learning (Cabannes et al., 2023a; Sobal et al., 2024), as even dissimilar pairs may share semantic attributes (Tsai et al., 2021). Hierarchical representation learning solves this problem by using a semantic hierarchy to capture the shared attributes between different classes (Ridnik et al., 2021; Mohammed & Umaashankar, 2018). Positive Active Learning (PAL) (Cabannes et al., 2023a) reveals that the learning process of both SL and SSL can be expressed in terms of a similarity graph. $\mathbb{X}$-CLR (Sobal et al., 2024) utilizes the text caption to facilitate the calculation of the similarity graph. Although these work has begun to explore more complex relationships between samples, the requirements of additional annotation or labelling limit their applications. Labels in SimLAP serve a distinct purpose compared to traditional supervised learning methods like Liu et al. (2017); Morin & Bengio (2005); Murtagh & Contreras (2012). Instead of assigning each sample to a single corresponding cluster (Liu et al., 2017), our approach uses labels to select relevant features.

**Common Features across Classes.** Some contrastive learning (CL) methods extract common features across classes by using extra variables, such as auxiliary information (Tsai et al., 2021) and attributes (Ma et al., 2021). Nevertheless, the above methods do not address the challenge of learning from distinct samples, e.g. snake-lamp. For example, hierarchical classification methods (Deng et al., 2014; Yan et al., 2015) utilize a predefined hierarchy to capture the shared attributes between different classes. However, constructing such a hierarchy requires significant human effort and domain knowledge. Essentially, these methods still rely on human knowledge to define the relationships between classes. Cross-class features have been shown to mitigate the collapse of decision boundaries (Kim et al., 2024), yet this exploration remains limited in incremental learning. Although common features across samples bring insights on new pretext tasks, such as cross-view completion (Weinzaepfel et al., 2022) and matryoshka representation (Kusupati et al., 2022), introducing this principle into CL is challenging.

## 3 ARBITRARY PAIR LEARNING

We introduce a novel learning paradigm that diverges from the conventional objective of maximising intra-group (i.e., class or instance) similarity while minimising inter-group similarity. Instead, we optimize similarity between arbitrary pairs of groups, without assuming any predefined semantic relationships or inter-class structure. *A key challenge is how to formulate the learning, particularly in the absence of supervision indicating which group pairs should be similar.* To address this, we introduce a subspace-based similarity optimisation framework, where learning is constrained to subspaces defined by selected group pairs. Let us dive into motivation of learning in subspaces.

### 3.1 MOTIVATION FOR LEARNING IN SUBSPACES

In the absence of explicit relational guidance, we initially regard every class pair as potentially positive. Yet, symmetrically attracting all pairs collapses the representation space into conflicting regions. Intuitively, we project representations into a specialized subspace to emerge shared attributes for an arbitrary pair. Before formulating our approach, we would like to investigate whether any similarity between arbitrary pairs in sub-spaces exists for current learning algorithms. To demonstrate such property, we chose a pretrained ResNet50 using supervised learning on IN1K from timm (Wightman, 2019). We extract the class vectors for all the samples in IN1K validation set corresponding to two distinct classes 'garter snake' and 'table lamp', which have no apparent common features. We visualize class similarity distribution in Figure 1 (b), see calculation in Appendix B.2 in Appendix. Our subspace selection, picking features that have the same sign, promotes similarity

for arbitrary pairs in selected subspaces. The two distinct classes are nearly orthogonal in the global (full) space, yet close in their discriminative subspace. This means that our learning can be achieved by learning/optimising for subspaces where the similarity of the corresponding arbitrary class pairs is maximised.

> **Observation:** For high-dimensional orthogonal representations, we can find a subspace for an arbitrary class pair such that their samples are close to each other in that subspace and far away from samples of other classes.

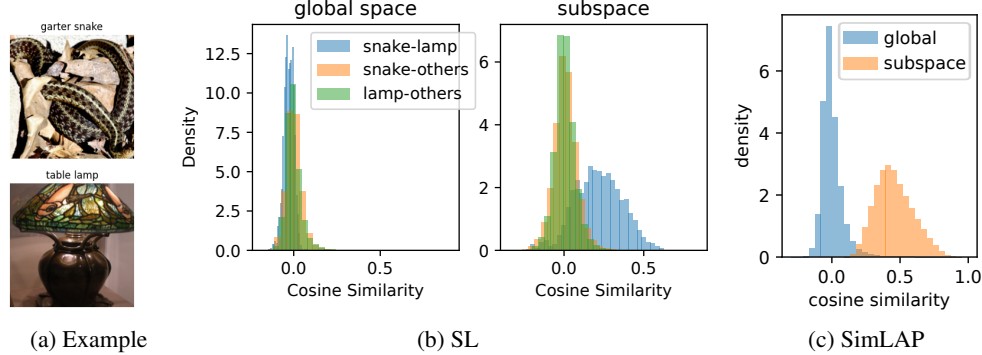

(a) Example            (b) SL              (c) SimLAP

Figure 1: Empirical evidence of discriminative subspaces for ResNet50 trained with SL. The class similarity distributions (Equation (8)) in global space and in the selected subspace shows that two distinct classes snake-lamp **(a)** are nearly orthogonal in global space but closer to each other than other classes in a subspace **(b)**. SimLAP enhances the similarity of lamp-snake in its subspace to around $0.48$ while preserving dissimilar in global space **(c)**.

## 3.2 CONTRASTIVE LEARNING

We revisit the basics of contrastive learning, which promotes the discovery of discriminative features between positive and negative samples. The key to contrastive learning is defining positives and negatives samples. For each anchor sample in the dataset $\mathbf{x} \in \mathcal{X}$, a conventional contrastive learning method defines its positives $\mathbf{x}^+$ and negatives $\mathbf{x}^-$ according to its learning principles. An encoder network $f_V(\cdot)$ is applied to map the images to a representation vector. A projection network $f_P$ is critical for improving transfer learning performance (Chen et al., 2020) and mitigating dimensional collapse (Jing et al., 2022). Overall, features are extracted by $\mathbf{z} = f_P(f_V(\mathbf{x}))$. The objective helps to identify the features that can separate positives and negatives, which is achieved by InfoNCE (Oord et al., 2018):

$$\mathcal{L} = -\log \frac{\exp(\langle \mathbf{z}, \mathbf{z}^+ \rangle / \tau)}{\exp(\langle \mathbf{z}, \mathbf{z}^+ \rangle / \tau) + \sum_{\mathbf{z}^- \in \mathbf{Z}^-} \exp(\langle \mathbf{z}, \mathbf{z}^- \rangle / \tau)}, \tag{1}$$

where $\langle u, v \rangle = \frac{u \cdot v}{\|u\| \|v\|}$ denotes the cosine similarity between two vectors.

## 3.3 LEARNING FROM ARBITRARY PAIRS

SimLAP maximises the similarity between two samples drawn from an arbitrary pair of classes, constrained to their corresponding subspace. In other words, the learning objective seeks subspaces where the similarity in terms of common features between selected arbitrary class pairs is maximised. The overall architecture is depicted in Figure 2. We randomly pair two samples $\mathbf{x}_i$ and $\mathbf{x}_j$ in a batch belonging to any class pair $y_i$-$y_j$ and select their corresponding subspace by an *adaptive feature filter*. The subspace can be sampled by shuffling cf. Appendix B.1. In this subspace, the samples from $y_i$-$y_j$ comprise positive pairs and the samples from any other classes comprise negative pairs. Specifically, for an anchor sample $\mathbf{x}_i$ from class $y_i$ in a mini-batch, we randomly choose one sample $\mathbf{x}_j$ from the mini-batch belonging to class $y_j$ and treat the sample belonging to $y_j$ as positive, $\mathbb{P}$. The samples pairs in the mini-batch that do not belong to $y_i$ and $y_j$ pair are negative $\mathbb{N}$. To maximise the similarity between arbitrary positive pair corresponding to anchor sample $\mathbf{x}_i$ and

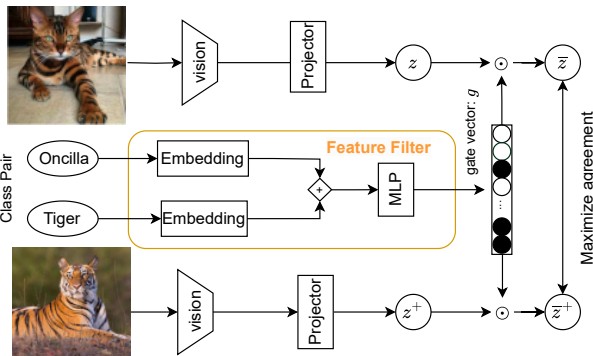

Figure 2: **Learning from Arbitrary Pairs**. SimLAP learns representations by maximizing the agreement of common features between arbitrary classes in a subspace, cf. Appendix B.1.

minimise the similarity between negative pairs we adapt infoNCE loss. The loss function for the anchor sample is formulated as follows: $\mathcal{L}(y_i, y_j) =$

$$- \sum_{\bar{z}^+ \in \mathbb{P}} \frac{1}{|\mathbb{P}|} \log \frac{\exp(\langle \bar{z}, \bar{z}^+ \rangle / \tau)}{\exp(\langle \bar{z}, \bar{z}^+ \rangle / \tau) + \sum_{\bar{z}^- \in \mathbb{N}} \exp(\langle \bar{z}, \bar{z}^- \rangle / \tau)} \quad (2)$$

where $\{\bar{z}, \bar{z}^+, \bar{z}^-\} = g(y_i, y_j) \odot \{z, z^+, z^-\}$ denote the features in the subspace of $y_i$-$y_j$. The elements of the gate vector $g(y_i, y_j)$, ranging from 0 to 1, control the activation of each feature.

Note that from a mini-batch size of 100 it is possible to build 5050 arbitrary positive pairs. However, for efficiency and for fairness to other learning methods we only consider 100 positive pairs where each sample in the mini-batch becomes anchor only once.

**Feature Filter.** The feature filter is a vital component to enable learning from arbitrary pairs. It selectively activates certain dimensions, effectively generating a subspace for the selected class pair, to represent their common features. This module enables the discovery and utilization of shared information between distinct samples. The feature filter is a simple lightweight module consisting of two parts: a label embedding and an MLP network. The label embedding converts discrete labels to continuous vectors. We use the mean of the two label vectors to represent the common information between two classes. The MLP layer generates gate vectors to select the corresponding dimensions for the common information. The calculation is

$$g(y_i, y_j) = \sigma(f_g((f_l(y_i) + f_l(y_j))/2)), \quad (3)$$

where $f_l(\cdot)$ denotes an embedding layer that converts a label to a 512-dimensional vector, $f_g(\cdot)$ denotes MLP network consisting of three linear layers with ReLU non-linearity in between them, and $\sigma(\cdot)$ is the Sigmoid activation function. In this framework, each class is equally pulled to or pushed from other classes in different subspaces, meaning that labels do not promote compact embeddings explicitly for any specific class pair.

All model parameters are randomly initialized to evaluate SimLAP capacity to learn from arbitrary visual pairs using only discrete labels. While label embeddings could be initialized from a pretrained LLM to leverage semantic priors, including potential similarities between arbitrary class pairs, we intentionally avoid this to isolate the learning ability of SimLAP without external knowledge. We use label indices, symmetric pairing, random initialized parameters to avoid introducing biases from labels. There is no explicit guidance on similarity between classes, e.g, in terms of detailed text giving common attributes etc. The model must learn it from visual information. Intriguingly, SimLAP learns the similarity between classes merely from visual information, cf. Figure 3. Our analysis reveals that the gate vectors will learn to select the dimensions with the same sign cf. Appendix A.2. More details can be found in Appendix B.1 with an algorithm outlined in Algorithm 1.

Table 1: **Transfer learning performance with KNN (k=10)** across 8 classification tasks. See full comparison in Appendix Table 9.

| top-1 | Aircraft | CF10 | Cars | DTD | FLW | Food | Pets | STL | AVG |
|---|---|---|---|---|---|---|---|---|---|
| SL | 28.5 | **88.0** | 31.6 | 66.5 | 68.3 | **58.4** | **91.1** | **97.3** | 66.2 |
| Supcon* | 33.7 | 83.8 | 30.6 | 55.0 | 68.3 | 53.5 | 88.8 | 95.4 | 63.6 |
| SimCLR* | 21.2 | 85.3 | 15.2 | **67.6** | 66.9 | 51.1 | 71.7 | 93.4 | 59.0 |
| SimLAP | **40.5** | 85.3 | **34.8** | 62.0 | **74.9** | 55.3 | 89.2 | 95.4 | **67.2** |

## 4 EXPERIMENTS

### 4.1 EXPERIMENTAL SETTING

We trained SimLAP on ImageNet (Deng et al., 2009) where its 0.5M subspaces are optimized efficiently in only 1M iterations. We choose two baseline methods to compare the effectiveness of arbitrary pairs compared to identical pairs. The backbone is ResNet50 (He et al., 2016). SL denotes the standard supervised setting with cross-entropy and one-hot vectors. We use the pretrained ResNet-50 from timm (Wightman, 2019). We trained Supcon (Khosla et al., 2020) as described in Appendix B.3. These two methods represent the majority in utilizing labels to promote the compactness of identical pairs. Note that our implementation of Supcon* is not as good as reported in the original paper due to a small batch size (1024) and a long-time training (1000 epochs), because the contrastive framework is sensitive to batch size (Chen et al., 2020). We apply the same training settings for both Supcon* and SimLAP to fairly compare the efficiency of learning from identical pairs and arbitrary pairs. The implementation in this section follows a minimal design to quantify the value of arbitrary pair in comparison to identical pairs. Appendix C demonstrates more evaluations as well as an improved version with a momentum encoder.

**Embedding Analysis.** We visualize the learned embeddings from the backbone network (excluding the projector and filter) by analysing representations on the IN1K validation set. As shown in Figure 5, SimLAP exhibits good statistical character in class separation. We also employ $t$-SNE (Van der Maaten & Hinton, 2008) to visualize the embeddings of 10 classes from the IN1K validation set in Figure 3. The visualization reveals two key findings: (1) In the global space, SimLAP maintains clear class separation as good as the methods optimized in the global space, despite being optimized in subspaces; (2) When examining the subspace for specific class pairs (e.g., Garter snake-Chihuahua), our model successfully identifies shared features that bring these seemingly distinct classes closer while preserving the overall structure of the embedding space.

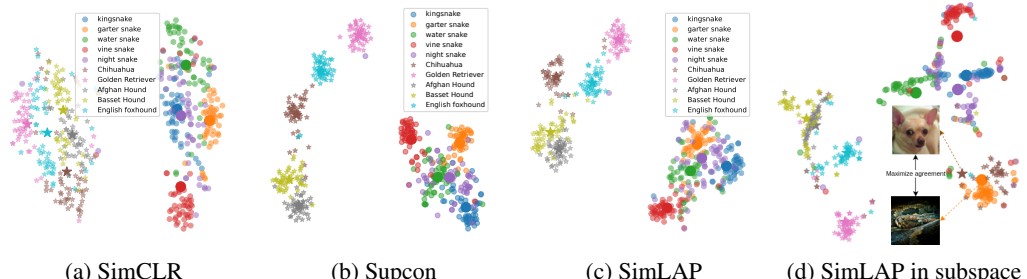

| (a) SimCLR | (b) Supcon | (c) SimLAP | (d) SimLAP in subspace |

Figure 3: $t$-**SNE visualization** of learned embeddings for 10 classes from two groups: 5 dog breeds (star) and 5 snake species (dot). Large markers indicate class centers, with stars representing dog classes and points representing snake classes. All models separate coarse classes (dogs and snakes). SimCLR struggles to differentiate between dog breeds. (d) highlights the unique ability of SimLAP to bring Chihuahua and garter snake close together within a subspace. Interestingly, optimizing disparate pairs in subspaces results in similar global embedding to supervised one (Supcon).

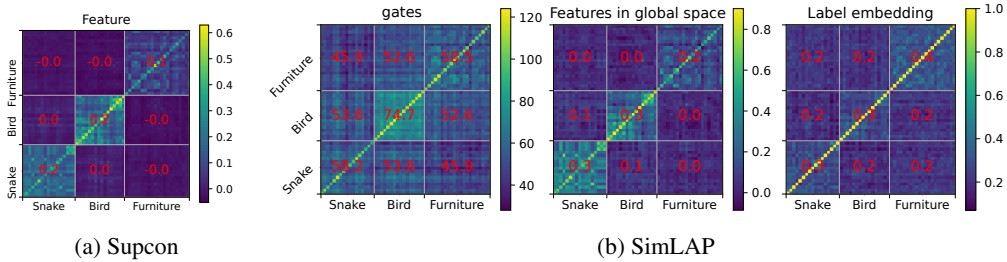

(a) Supcon                                                    (b) SimLAP

Figure 4: **Class similarity measured by gates, global features, and label vectors.** Red numbers denote the average similarity within each super synset region (snake, wading bird, furniture). Higher class similarity of super synset indicates that feature filter in SimLAP learns the relationships between classes from representations.

Table 2: Image-retrieval results (mAP) on ROxford5k and RParis6k (Radenović et al., 2018).

|  | Medium (M) | | | Hard (H) | | |
|---|---|---|---|---|---|---|
|  | SL | Supcon* | SimLAP | SL | Supcon* | SimLAP |
| ROxford5k | 27.13 | 26.28 | **30.82** | 9.22 | 6.6 | **9.48** |
| RParis6k | 53.41 | 49.35 | **54.29** | **28.62** | 24.8 | 28.31 |

## 4.2 TRANSFER LEARNING

Although our framework optimizes representations in subspaces, global representations exhibit good class separation and transfer learning performance. See more evaluations in Appendix C.

**Image Classification with K-nearest Neighbour (KNN).** We use the KNN protocol to present the transfer learning performance. KNN can largely reveal the learned semantic measurement. As shown in Table 1, SimLAP exhibits competitive even superior performance on downstream tasks. This indicates that distinct pairs could be as valuable as identical pairs.

**Image Retrieval.** Retrieval is an important application of representation learning to find images for a query image (Radenović et al., 2018). As shown in Table 2, SimLAP consistently outperforms two baselines. These results confirm that the intra-class orthogonality encouraged by SimLAP produces more discriminative global descriptors, leading to superior image-search quality without any task-specific re-ranking or query expansion.

**Dense Prediction.** Classification tasks reflect the quality of global features. To further study the ability to extract subtle information in local representations, we performed semantic segmentation with frozen features. As shown in Table 3, SimLAP extract meaningful local characteristics. The property can be further improved by introducing momentum encoder, see Table 10.

## 5 MODEL ANALYSIS

> **Finding 1:** Filter makes learning robust to distinct pairs.

The major difficulty of SimLAP is dealing with distinct pairs in which common visual similarity can hardly be found, such as snake-lamp. To study the impact of reducing common features in pairs, we conducted a controllable experiment in increasing the semantic distance of positive pairs. We run the experiment on IN25, a dataset of 25 classes with well-defined semantic relationships (details in Appendix B.3). Each model was trained for 1000 epochs. As shown in Figure 6, the performance of the model without the filter decreases over semantic distance, because distinct pairs define a wrong semantic relationship between classes. In contrast, the model with filter is robust in dealing with distinct pairs.

> **Finding 2:** SimLAP simultaneously optimises representations from visual data and subspaces for class pairs.

Figure 5: **Inter- and intra- Similarity distribution** on IN1K. The number in the middle denotes the overlap of two distributions. Although SimLAP only optimizes representations in subspaces, the representations in global space exhibit class separation (small overlap) like SL instead of SSL. See calculation in Equation (8).

Figure 6: **Adaptive filter is crucial to deal with distinct pairs.** X axis denote semantic distance of the positive set (ranked by CLIP). Models can hardly learn meaningful representations from distinct pairs without the feature filter.

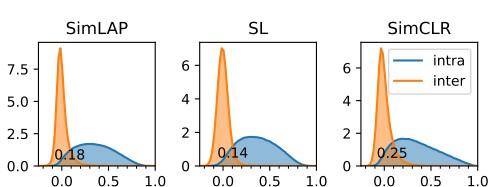

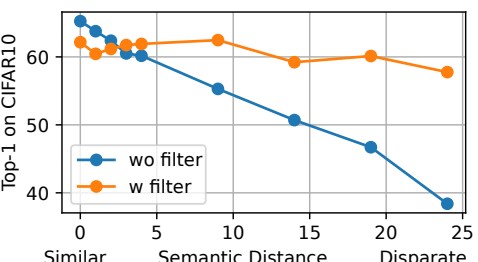

Table 3: **Segmentation** with frozen features via FCN (Long et al., 2015).

| mIoU | cityscapes | voc12aug |
|---|---|---|
| SL | 52.14 | **54.9** |
| Supcon* | 50.50 | 43.8 |
| SimCLR* | **54.5** | 44.8 |
| SimLAP | 49.92 | 45.76 |

Table 4: **Optimizing multiple subspaces in a step.** Runtime (seconds) measures the elapse time to forward 128 samples, tested on a single V100. Report top-1 with KNN on CF10.

| Num | GFlops | Runtime | act. | CF10 |
|---|---|---|---|---|
| 1 | 1,063 | 0.30 | 63 | 85.14 |
| 5 | 1,064 | 0.32 | 53 | 85.96 |
| 20 | 1,067 | 0.39 | 28 | **86.21** |

Our model can be divided into two independent learning processes: identifying the subspace for each class pair and learning the common features of the class pair. By default, we randomly select a single subspace to optimize these representations. We investigate the impact of increasing the number of optimized subspaces. Specifically, we first compute global features by forwarding images once, then generate multiple subspaces using Equation (3) to optimize the objective in Equation (2). As shown in Table 4, increasing the number of optimized subspaces enhances performance but increases training time. For simplicity and computational efficiency, we default to optimizing a single subspace for each training step.

The feature filter captures class similarity in gate vectors and label embedding as shown in Figure 4. SimLAP is able to automatically detect the visual biases in images (spatial locality, transformation invariance, and shared attributes across classes) and inherit these properties by the filter.

> **Finding 3:** The common features across classes are effective in preventing collapse.

To thoroughly investigate the collapse issues, we construct IN1P, a focused subset of IN1K containing 12K images from 10 highly-related dog breeds. This small dataset allows us to train extensively on all possible class pairs. We train models for 20,000 epochs to ensure that models reach minimum. In Figure 7a, we compare the top-1 accuracy with KNN on CIFAR10 across different methods. SimCLR exhibits clear signs of dimensional collapse (Jing et al., 2022) after 3,000 epochs, evidenced by the declining performance and decreasing singular values, from Figure 7b. This reveals that while the non-linear projector can relieve dimensional collapse, it cannot prevent it entirely. Incorporating supervision into CL is a promising solution (Xue et al., 2023). SimLAP demonstrates sustained performance improvement throughout the extended training period, surpassing both Supcon and SimCLR. This robustness against dimensional collapse and class collapse can be attributed to discovering rich common features between classes.

## 6 APPLICATION

Our learning scheme is not a replacement of supervised learning but a complementary one to enrich representations with cross-class features. We demonstrate two ways of combining SimLAP with self-supervised learning algorithms.

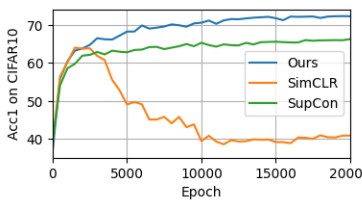 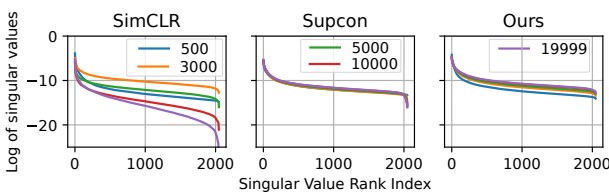

(a) Performance during training   (b) Singular value spectrum of the embedding space

Figure 7: **Preventing collapse by learning common features**. Lower singular values suggest that representations are concentrating information in fewer dimensions. Models trained on IN1P (10 dogs) and evaluated on CIFAR10 show that SimLAP consistently outperforms baseline methods.

**Joint Loss.** Our learning paradigm is complementary to existing ones, bringing further improvements. To combine SimLAP and MoCov3 (He et al., 2020), we introduce a joint loss by $\frac{1}{1+\beta}\mathcal{L}_{\text{SSL}} + \frac{\beta}{1+\beta}\mathcal{L}_{\text{SimLAP}}$. From Table 5, the performance is improved by balancing two losses.

**Mid-training.** Mid-training is a stage between SSL (MAE (He et al., 2022) here) and supervised finetuning (SFT). From Table 6, we can see a significant performance drop for long SFT due to over-adaptation (Hao et al., 2025). SimLAP relieves this problem and gains 0.4% improvement.

Table 5: Joint loss of SimLAP and MoCov3 (ResNet50) improves representations, cf. Appendix B.3.

| beta | CIFAR10 | DTD | FLW |
|------|---------|-------|-------|
| 0 | 82.90 | 63.35 | 83.10 |
| 1 | **90.33** | **64.31** | **88.18** |
| 100 | 87.72 | 58.51 | 85.23 |

Table 6: SimLAP (ViT-Base) relieves over-adaption, cf. Appendix B.3. FT and SimLAP train 100 epochs. FT* trains 200 epochs.

| training pipeline | top1 |
|-------------------|------|
| SSL (1600) $\rightarrow$ FT (100) | 83.6 |
| SSL (1600) $\rightarrow$ FT* (200) | 83.1 |
| SSL (1600) $\rightarrow$ SimLAP (100) $\rightarrow$ FT (100) | **84.0** |

# 7 DISCUSSION

Our study demonstrates that learning from arbitrary pairs is not only feasible but also beneficial. By introducing SimLAP, we show that even pairs of semantically distant classes can reveal common features when constrained to appropriate subspaces. This challenges the conventional "one-vs-all" paradigm and highlights arbitrary pairs as a complementary source of supervision. The framework consistently uncovers shared subspaces while preserving class separability, leading to competitive transferability and robustness against collapse. To keep the focus on this principle, we adopted a minimal contrastive setup, avoiding additional components, e.g. momentum encoder. While such techniques could further boost performance, our goal was to isolate the effect of arbitrary pairs and make their contribution explicit. The results suggest that arbitrary-pair supervision is especially valuable when identical pairs suffer from a diminished signal due to overfitting.

**Limitation.** Given the large number of possible pairs ($\approx 0.5$M in IN1K), our 1M iteration training could not fully explore the available space, leaving many informative relations not fully leveraged. Further, not all pairs are meaningful, e.g., "snake-lamp" pair lack obvious commonalities. How to efficiently identify and utilise the valuable pairs without prior annotation remains an open challenge.

**Future Work.** A promising direction is to extend arbitrary-pair learning with richer forms of supervision. In multimodal data, for example, two images with captions such as "a clock is mounted to a house" and "a clock beside a brick sidewalk on the Baker Street" both highlight the concept of "clock". Feature filters could exploit this overlap to align disparate visual inputs. Incorporating textual cues, attributes, or hierarchical metadata may further enhance the robustness and transferability of learned representations.

## 8 ETHICS STATEMENT

We study a generic mechanism for learning representations from arbitrary sample pairs. No private or sensitive information was collected or processed. All experiments rely on publicly available datasets (ImageNet-1K, CIFAR, etc.) or standard splits derived from them. We are not aware of any foreseeable privacy breaches or discriminatory consequences arising from this work.

## 9 REPRODUCIBILITY STATEMENT

Implementation details, hyper-parameters, and training recipes are documented in Appendix B. Source code and pretrained weights are provided in the supplementary material. All datasets except IN1P and IN25 are publicly downloadable; the latter two can be constructed from ImageNet-1K with the scripts we supply, ensuring full reproducibility. The training code of SimLAP on CIFAR10 is provided.

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

# Appendix

## A    THEORY PROOF

### A.1    EXISTENCE OF DISCRIMINATIVE SUBSPACES

Our work is based on the hypothesis that there exists a subspace to allow arbitrary pairs to be close while the other classes are far away. In general, two orthogonal class vectors decompose into a negative and a positive component whose sum is zero. The positive component, the same sign in every dimension, generates a subspace in which similarity is encouraged.

We ground this selection strategy in the realistic assumption that the representations follow a Gaussian distribution. Under this assumption we prove that, for any pair, the expected similarity inside the positive subspace is $1/\pi$ when the pair is drawn from the same class and $0$ in global space. An artificial example Figure 8 supports our theory in Equation (4). It indicates that our idea of selecting a subspace for arbitrary pairs is is model-agnostic and applies equally to supervised and self-supervised models.

**Definition.** Let N orthogonal class vectors be sampled from D-dimensional Gaussian distributions: $\boldsymbol{z} \in \mathbb{R}^{N \times D}, \boldsymbol{z}_i^d \overset{\text{i.i.d.}}{\sim} \mathcal{N}(0, 1/D)$.

**Subspace Selection.** For an arbitrary pair $(i, j) \in \mathbb{P}^+$, we select the gate vector $\boldsymbol{g}(i, j) = I_d = \mathbb{1}_{\boldsymbol{z}_i^d \boldsymbol{z}_j^d > 0}$. The chance of each dimension being positive is 0.5: $p(\boldsymbol{g}(i,j) = 1) = p(\boldsymbol{z}_i^d > 0)p(\boldsymbol{z}_j^d > 0) + p(\boldsymbol{z}_i^d < 0)p(\boldsymbol{z}_j^d < 0)$. Hence $\boldsymbol{g}(i,j) \sim \text{Binomial}(D, 0.5)$.

For orthogonal representations, given an arbitrary pair, we can select a subspace with positive sign and have the following:

$$\begin{aligned} \mathbb{E}_{(i,j) \in \mathbb{P}^+}[\bar{\boldsymbol{z}}_i \bar{\boldsymbol{z}}_j] &= \frac{1}{\pi}, \\ \mathbb{E}_{(i,k) \in \mathbb{P}^-}[\bar{\boldsymbol{z}}_i \bar{\boldsymbol{z}}_k] &= 0, \end{aligned} \tag{4}$$

where $\bar{\boldsymbol{z}} = \boldsymbol{z} \odot \boldsymbol{g}(i, j)$.

**Proof.** Fix a positive pair $(i, j) \in \mathbb{P}^+$ and write

$$\boldsymbol{g} = \boldsymbol{g}(i, j) \in \{0, 1\}^D, \qquad \boldsymbol{g}_d = \mathbf{1}_{z_i^d z_j^d > 0}.$$

Because $z_i^d, z_j^d \overset{\text{i.i.d.}}{\sim} \mathcal{N}(0, 1/D)$, the product equals to only consider the half part. The coordinate-wise conditional expectation of the product given the gate is

$$\mathbb{E}\big[z_i^d z_j^d \mid \boldsymbol{g}_d = 1\big] = \mathbb{E}[|z_i^d||z_j^d|] = \frac{2}{\pi}\frac{1}{D},$$

because the mean of a standard half-normal is $\mathbb{E}[|X|] = \sqrt{\frac{2}{\pi}}, X \sim \mathcal{N}(0, 1)$.

In addition,

$$\Pr(\boldsymbol{g}_d = 1) = \Pr(z_j^d > 0) = \tfrac{1}{2} \quad \text{and} \quad \boldsymbol{g}_d \overset{\text{i.i.d.}}{\sim} \text{Bernoulli}(1/2).$$

Hence

$$\mathbb{E}_{(i,j) \in \mathbb{P}^+}\big[\boldsymbol{g}^\top (z_i \odot z_j)\big] = \sum_{d=1}^{D} \mathbb{E}\big[\boldsymbol{g}_d z_i^d z_j^d\big] = \sum_{d=1}^{D} \Pr(\boldsymbol{g}_d = 1)\,\mathbb{E}\big[z_i^d z_j^d \mid \boldsymbol{g}_d = 1\big] = D\frac{2}{\pi}\frac{1}{D}\frac{1}{2} = \frac{1}{\pi}.$$

For a negative pair $(i, k) \in \mathbb{P}^-$ the variables $z_i^d$ and $z_k^d$ are independent and zero-mean, so

$$\mathbb{E}\big[z_i^d z_k^d\big] = 0 \quad \text{regardless of the gate.}$$

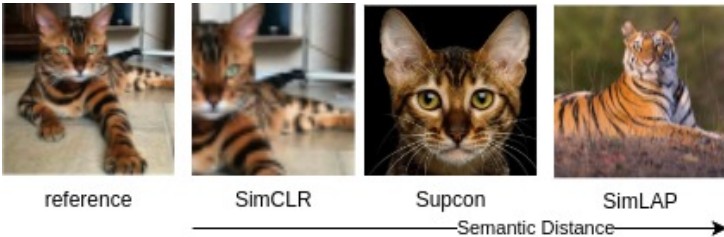

Figure 8: Comparison of sample pairs in three CL methods.

**Stimulation.** We simulate the process of feature selection by positive contribution. We generate 1000 samples from 10 class vectors, $\mathbf{x} = \mathbf{y} + \epsilon\mathbf{e}$, where $\mathbf{y} \in \mathcal{N}(0, 1/D), \epsilon = 0.0001$, D denotes the size of representations. The gap is defined as the difference between the macro-class similarity and the largest similarity observed to any negative sample; a larger gap indicates sharper representation separability:

$$\text{gap}(i, j) = \boldsymbol{z}_i\boldsymbol{z}_j - \max_k \boldsymbol{z}_i\boldsymbol{z}_k. \tag{5}$$

Our simulated results match our theoretical analysis, as shown in Figure 9. The results corroborate our theoretical claim: for any two class vectors there exists a subspace in which their similarity is selectively amplified.

## A.2 GRADIENT ANALYSIS

Consider a simplified situation, where $C$ samples are drawn from $C$ distinct classes, denoted $(x_i, y_i)$. The pairwise similarity score between samples $i$ and $j$ is defined as $s(i, j) = \langle \mathbf{g} \odot z_i, \mathbf{g} \odot z_j \rangle / \tau$, parameterized by a gate vector $\mathbf{g} = g(y_i, \bar{y}_j) \in \mathbb{R}^D$, $\bar{y}$ is the shuffled labels. For a subspace of class pair $y_i$-$y_j$ and temperature $\tau$. The objective function $L$ minimizes the InfoNCE loss for two positive pairs:

$$L = -\log\left(\frac{\exp(s(i, j))}{\sum_{k \neq j} \exp(s(i, k))}\right) - \log\left(\frac{\exp(s(j, i))}{\sum_{k \neq i} \exp(s(j, k))}\right). \tag{6}$$

To optimize $L$, the gradient of $L$ with respect to each dimension $d$ of the gate vector $\mathbf{g}$ is derived as:

$$\frac{\partial L}{\partial g^d} = \frac{2g^d}{\tau}\Big[ \sum_{k \neq i, j} p(x_k|i)z_i^d z_j^d - (1 - p(x_j|i))z_i^d z_J^d$$
$$\sum_{k \neq i, j} p(x_k|j)z_j^d z_j^d - (1 - p(x_i|j))z_i^d z_j^d\Big]. \tag{7}$$

Here, $p(x_k|i) = \frac{\exp(s(i,k))}{\sum_{j \neq i} \exp(s(i,j))}$ denotes the softmax probability of sample $k$ being the class $i$. The sign of the gradient is governed by the product $z_i^d z_j^d$, with its magnitude adjusted by $\mathbf{g} \in [0, 1]^D$. Specifically, a positive product between positive pairs ($z_i^d z_j^d > 0$) or a negative product between negative pairs ($z_I^d z_j^d < 0$) promotes an increase in $g^d$. The gate vectors will be nearly binary to find the subspace for an arbitrary pair by activating or inactivating dimensions cf. Figure 15.

## B  IMPLEMENTATION DETAIL

### B.1  CONTRASTIVE LEARNING WITH ARBITRARY PAIRS

Another way of looking at SimLAP is from the prospective contrastive learning. SimLAP can be seen as extending contrastive learning from similar sample pairs to arbitrary pairs. Figure 8 compares the positive pairs in SimCLR, Supcon, and SimLAP. Our algorithm selects positive and negative pairs according to the generated subspace as shown in Figure 10. Algorithm 1 outlines our framework for learning from arbitrary pairs. The algorithm begins by sampling mini-batches of input data and their corresponding labels. Notably, we sample positive pairs from the mini-batch instead of augmented views. For each pair of classes $y_{2k-1}$-$y_{2k}$, it generates a subspace

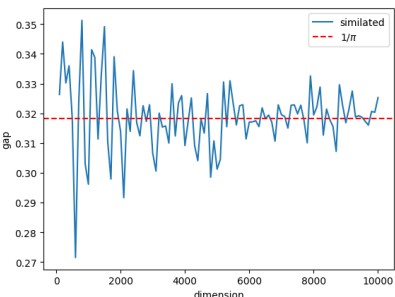

Figure 9: The expectation of the gap between intra-class and inter-class fits the theoretical value.

| subspace | label | y2 | y1 | y3 | y3 | y2 |
|----------|-------|----|----|----|----|----|
| y2-y2 | y2 | p | n | n | n | p |
| y1-y2 | y1 | p | p | n | n | p |
| y3-y1 | y3 | n | p | p | p | n |
| y3-y3 | y3 | n | n | p | p | n |
| y2-y3 | y2 | p | n | p | p | p |

Figure 10: Positive and negative samples in SimLAP. 'p' denotes positive, 'n' denotes negative.

representation using the gate function $f_g$, which takes the average of the label embeddings as input. The encoder $f_V$ extracts representations for both the reference sample and its arbitrary pair. These representations are then projected into the global space using the projector $f_P$ and element-wise multiplication with the gate vector $g_k$ to project into subspaces. Pairwise similarities are computed for all samples in the batch using cosine similarity. The contrastive loss is calculated using these similarities and a temperature parameter $\tau$. This loss encourages the model to maximize similarity between arbitrary pairs in their shared subspace while minimizing similarity with samples from other classes. The networks $f_V$, $f_P$, $f_l$, and $f_g$ are updated to minimize this loss. After training, only the encoder network $f_V(\cdot)$ is retained for downstream tasks, discarding the gating mechanism used during training. This approach allows the model to learn transferable representations that capture common features across seemingly disparate classes.

**Subspace Sampling.** Each sample can be optimized within multiple related subspaces. Ideally, we uniformly sample one subspace from all available options. However, when the batch size is small, there is a risk of sampling a subspace without positive pairs in the batch. To mitigate this, we can resample until a non-empty subspace is obtained. Alternatively, we employ a simpler strategy by shuffling the labels within a batch. This ensures that at least one positive pair is present in the sampled subspace.

## B.2 CLASS SIMILARITY

There are two types of class similarity calculation: micro-class similarity and macro-class similarity. Micro-class similarity is a measure that focuses on the similarity between individual instances within two classes, capturing the relationships between the elements from the class pair. Specifically, the micro-class similarity between two classes $y_1$ and $y_2$ is defined as the expected value of the cosine distance $\langle \cdot, \cdot \rangle$ between pairs of instances from these classes:

$$\text{Micro-Sim}(y_1, y_2) = \mathbb{E}[\langle z_i, z_j \rangle], \quad z_i \in y_1, \ z_j \in y_2. \tag{8}$$

This formulation ensures that the similarity is computed directly from the pairwise interactions of instances, providing a detailed and granular measure of how closely the elements of the two classes are related.

In contrast, macro-class similarity provides a more aggregate measure by comparing the expected values (or means) of the instances within each class. This approach captures the overall similarity between the central tendencies of the classes, rather than focusing on individual instances. The macro-class similarity between $y_1$ and $y_2$ is defined as:

$$\text{Macro-Sim}(y_1, y_2) = \langle E[z_i], E[z_j] \rangle, \quad z_i \in y_1, \ z_j \in y_2 \tag{9}$$

Here, $E[z_i]$ and $E[z_j]$ represent the expected values (or means) of the instances in classes $y_1$ and $y_2$, respectively. By comparing these expected values, macro-class similarity offers a higher-level view of the relationship between the classes, which can be particularly useful when the overall trends or central tendencies are of primary interest.

---

**Algorithm 1** SimLAP algorithm.

---

**input:** batch size $N$, constant $\tau$, structure of $f_V$ and $f_P$ for encoding, structure of $f_l$ and $f_g$ for the feature filter.

**for** sampled minibatch $\{\boldsymbol{x}_k\}_{k=1}^N, \{y_k\}_{k=1}^N$ **do**

   $\bar{y} = \text{shuffle}(\{y_k\}_{k=1}^{2N})$   # Compose random class pairs.

   **for all** $k \in \{1, \ldots, N\}$ **do**

      $\boldsymbol{h}_k = f_V(\boldsymbol{x}_k)$        # representation

      $\boldsymbol{z}_k = f_P(\boldsymbol{h}_k)$     # features in global space

   **end for**

   **for all** $i \in \{1, \ldots, N\}$ and $j \in \{1, \ldots, N\}$ **do**

      $\boldsymbol{g}_i = f_g((f_l(y_i) + f_l(\bar{y}_i))/2)$   # generate gate vectors.

      $\bar{\boldsymbol{z}}_i = \boldsymbol{z}_i \odot \boldsymbol{g}_i$     # features in subspace

      $\bar{s}_{i,j} = \bar{\boldsymbol{z}}_i^\top \bar{\boldsymbol{z}}_j / (\|\bar{\boldsymbol{z}}_i\|\|\bar{\boldsymbol{z}}_j\|)$       # pairwise similarity

   **end for**

   # Samples are from $y_k$ or $\bar{y}_k$ are positive.

   # Samples are not from $y_i$ or $\bar{y}_i$ are negative.

   **define** $\ell(i, j)$ **as**

   $\ell(i) = -\frac{1}{|\mathbb{P}|} \sum_{y_p \in \mathbb{P}} \log \frac{\exp(s_{i,p}/\tau)}{\exp(s_{i,p}/\tau) + \sum_{k \in \mathbb{N}} \exp(s_{i,k}/\tau)}$

       where $\mathbb{P}$ is the positive set, $\mathbb{N}$ is the negative set.

   $\mathcal{L} = \frac{1}{N} \sum_{k=1}^N \ell(k)$

   update networks $f_V, f_P, f_l,$ and $f_g$ to minimize $\mathcal{L}$

**end for**

**return** encoder network $f_V(\cdot)$, and throw away others

---

Table 7: **Pretraining details**. The output size of Vision Encoder depends on the backbone (2048 for ResNet50). The optimizer is LARS. Learning rate is scheduled by a cosine function with warmup.

(a) **Hyperparameter**.

|  | SimLAP | SimLAP† |
|---|---|---|
| Batch size | 1024 | 1024 |
| Epochs | 1000 | 300 |
| Warmup | 40 | 40 |
| LR | 0.2 | 0.3 |
| Weight decay | 1e-4 | 1.5e-6 |
| Runtime (h) | 56.3 | 25.3 |

(b) **Projector Structure**.

| Layer | Output |
|---|---|
| Vision Encoder | - |
| BatchNorm1d, ReLU | 2048 |
| Linear | 2048 |
| BatchNorm1d, ReLU | 2048 |
| Linear | 256 |
| LayerNorm | 256 |

(c) **Filter Structure**.

| Layer | Output |
|---|---|
| Label Embedding | 512 |
| BatchNorm1d, ReLU | 512 |
| Linear | 1024 |
| BatchNorm1d, ReLU | 1024 |
| Linear | 256 |
| Sigmoid | 256 |

## B.3 TRAINING DETAIL

**IN1K Pretraining.** We conducted extensive experiments on the ImageNet-1K (IN1K) dataset. Our experimental settings are summarized in Table 7a. The data augmentation techniques utilized in this work include random horizontal flipping, color jittering (brightness=0.4, contrast=0.4, saturation=0.4, hue=0.1), normalization to ImageNet statistics, random grayscaling (p=0.2), and Gaussian blurring. To accelerate the training procedure, we apply ESSL (Wu et al., 2024) and amp with half-precision. Experiments are conducted on a single machine with 8 Tesla V100-SXM2-32GB-LSs, 512G Mem, and 2 E5-2698 (40 cores). SimLAP† applies the momentum encoder (m=0.996) to enhance performance.

**IN1P Pretraining.** The IN1K pretraining comprises 0.5M pairs, and our implementation cannot exploit every pair. To evaluate a well-trained SimLAP, we consider a small dataset IN1P Appendix E to study how to effectively use class similarity. The model was trained on IN1P for 10,000 epochs with a batch size of 512. The backbone is ResNet50.

**IN25 Pretraining.** We conduct experiments on IN25 to study the impact of distinct pairs in representation learning. To control the semantic distance of sample pairs, we first calculate the macro-class similarity using CLIP (ViT-B/16). Then, we comprise positive sets according to the rank of class similarity.

**Joint Loss.** Our learning paradigm extracts different aspects of features from existing learning paradigms, thus is complementary to them. The simplest way to harvest both is to train with two heads in parallel: one governed by SimLAP, the other by MoCov3 (Chen et al., 2021). Both heads share the same projector architecture and receive identical augmented views; their losses are added with equal weight and back-propagated through the common backbone.

**Mid-training.** The classic 'pretraining-fientuning' pipeline yields state-of-the-art results, yet longer fine-tuning epochs inevitably tilt the model toward the downstream task and erode the generic representations inherited from pretraining. The performance will drop for lone-time finetuning as a consequence of over-adaptation.

We insert SimLAP as a bridging phase between the two stages to cushion this drift. Concretely, starting from an ImageNet-1K pre-trained checkpoint, we continue a pretrained model with SimLAP for 100 epochs (batch size 1 024, cosine LR schedule). This brief mid-training aligns knowledge without exposing the network to downstream labels, leaving the backbone better balanced for subsequent fine-tuning: generic enough to retain prior knowledge, yet discriminative enough to converge faster and plateau higher.

**Transfer Learning via Linear Evaluation.** A linear classifier was trained on features extracted from a pre-trained neural network. The L-BFGS optimization algorithm was utilized to minimize the softmax cross-entropy loss function. No data augmentation techniques were applied, aside from random cropping and resizing images to 224x224 pixels. No regularization methods, such as weight decay, were employed.

**Transfer Learning via Finetuning.** A linear classifier was attached to the vision backbone. The AdamW optimization algorithm was utilized to minimize the objective. The learning rate was dynamically changed by the cosine schedule. The maximum learning rate was determined by the base learning rate (following 1e-4 × BatchSize/256). Several regularization techniques were applied to prevent overfitting, including strong data augmentation, weight decay, and Mixup (Zhang, 2017). The data augmentation composes of RandCropResize(224, scale=(0.08,1)), RandomHorizontalFlip(p=0.5), AutoAugment(rand-m9-mstd0.5-inc1) (Cubuk et al., 2018), and RandErase(p=0.25) (Zhong et al., 2020). The images underwent mixup or cutmix (mixup_alpha=0.8, cutmix_alpha=1, label_smoothing=0.1). The weight decay was set to 2e-5. These settings may be not optimal, but we use the same settings for all datasets except the training epochs. We set training epochs to 90 for IN1K, 300 for iNat2018, 200 for Food, 1000 for the rest.

**Semantic Segmentation via Linear Evaluation.** MMSegmentation [1] was utilized to train FCNs (Long et al., 2015) for segmentation. Specifically, frozen features from four stages of ResNet50 are concatenated to represent local patterns. We applied three predefined configures (fcn_r50-d8_4xb2-40k_cityscapes-512x1024.py, fcn_r50-d8_4xb4-160k_ade20k-512x512.py, fcn_r50-d8_4xb4-20k_voc12aug-512x512.py) from MMSegmentation to train and evaluate our pretrained models.

## B.4    BASELINE MODEL

We use pretrained models from official repositories: MoCo (He et al., 2020)[2], DINO (Caron et al., 2021)[3], GPaCo (Cui et al., 2023)[4], BarlowTwins (BT) (Zbontar et al., 2021)[5], SimSiam (Chen & He, 2021)[6], SimCLR* (Chen et al., 2020)[7]. Supcon* (Khosla et al., 2020) were trained in the same settings as SimLAP for 1000 epochs (see Appendix B.3). SimCLR* and Supcon* are worse than these in the original papers due to aligning the training settings to ours.

---

[1] `https://github.com/open-mmlab/mmsegmentation`

[2] https://github.com/facebookresearch/moco

[3] https://github.com/facebookresearch/dino

[4] https://github.com/dvlab-research/Parametric-Contrastive-Learning

[5] https://github.com/facebookresearch/barlowtwins

[6] https://github.com/facebookresearch/simsiam

[7] https://github.com/facebookresearch/vissl

# C MORE EVALUATION

**Finetuning.** Pretraining-finetuning is a privilege way to leverage performance on various tasks. However, this protocol is not a good approach to evaluate the effectiveness of pretraining. The final results are largely effected by hyper-parameters. Training from scratch with a well-tuned recipe can also achieve a competitive result (He et al., 2019). Table 8 compares the transfer learning performance with finetuning protocol. SimLAP and SimLAP† are slightly worse than the SSL ones, but the gap is small.

Table 8: Comparison of transfer learning performance via **finetuning** across 8 classification tasks.

| top-1 | STL | CIFAR10 | Food | Aircraft | Cars | DTD | FLW | Pets | AVG |
|---|---|---|---|---|---|---|---|---|---|
| Supcon* | 97.7 | 95.0 | 83.2 | 78.5 | 93.0 | 70.1 | 91.1 | **92.2** | 87.6 |
| GPaCo | 97.9 | **98.3** | 89.1 | 89.3 | 92.8 | 70.4 | 94.1 | 92.2 | 90.5 |
| SimSiam | 94.9 | 96.7 | 88.7 | **91.1** | **93.7** | 68.5 | 92.2 | 90.1 | 89.5 |
| BT | **98.3** | **98.3** | **89.4** | 87.2 | 92.2 | 71.9 | 94.3 | 92.0 | **90.4** |
| SimCLR* | 97.7 | 97.9 | 89.1 | 89.8 | 93.0 | **72.4** | 95.3 | 91.1 | 90.8 |
| DINO | 97.9 | 97.9 | **90.3** | 88.6 | 93.2 | **73.1** | **95.7** | 91.7 | 91.0 |
| MoCo | 97.7 | 98.1 | 89.7 | **90.2** | **93.6** | 72.0 | 94.6 | **93.3** | **91.1** |
| SimLAP | 96.6 | 97.8 | 88.7 | 89.9 | 93.6 | 68.9 | **95.4** | 90.5 | 90.2 |
| SimLAP† | **98.0** | **98.3** | 88.7 | 89.3 | **93.6** | 71.8 | 94.5 | 93.2 | 90.9 |

**K-nearest Neighbors.** We compare SimLAP with other SL and SSL models across 8 classification tasks via KNN evaluation. The k-nearest-neighbours (KNN) algorithm classifies an unlabeled sample by finding the k closest labelled examples in feature space and assigning the majority class among them; Distance is measured by cosine similarity, and no explicit training phase is required. We report the top-1 accuracy with k=10 in Table 9. Momentum is an effective technique for improving transfer learning performance. For each category with or without momentum, our learning paradigm gains transferable features and surpasses both the SL and SSL models. Notably, SimLAP† gets the best results for 4 out of 8 tasks.

Table 9: Comparison of transfer learning performance via **KNN** across 8 classification tasks.

| top-1 | Aircraft | CF10 | Cars | DTD | FLW | Food | Pets | STL | AVG |
|---|---|---|---|---|---|---|---|---|---|
| SL | 28.47 | 88.02 | 31.55 | 66.54 | 68.33 | 58.38 | **91.11** | 97.3 | 66.21 |
| GPaCo | 20.07 | 89.58 | 23.38 | 61.12 | 52.28 | 48.61 | 90.05 | **97.96** | 60.38 |
| Supcon* | 33.66 | 83.79 | 30.62 | 55 | 68.32 | 53.52 | 88.83 | 95.41 | 63.64 |
| SimCLR* | 21.15 | 85.25 | 15.22 | **67.61** | 66.87 | 51.08 | 71.65 | 93.39 | 59.03 |
| BT | 33.66 | 87.61 | 25.54 | 68.88 | 80.11 | 60.42 | 82.53 | 94.81 | 66.70 |
| SimSiam | 27.93 | 86.68 | 18.13 | 63.94 | 74.29 | 52.27 | 72.64 | 93.21 | 61.14 |
| DINO | 37.53 | 88.79 | 30.43 | 70.69 | 82.44 | 61.2 | 81.41 | 95.61 | 68.51 |
| MoCo | 39.15 | **91.62** | 29.13 | 68.03 | 80.42 | 59.36 | 86.75 | 96.89 | 68.92 |
| SimLAP | 40.47 | 85.32 | 34.78 | 62.02 | 74.91 | 55.25 | 89.21 | 95.42 | 67.17 |
| SimLAP† | **42.54** | 91.52 | **39.14** | 65.85 | **81.17** | **61.37** | 90 | 97.2 | **71.10** |

**Dense Prediction.** Classification tasks reflect the quality of global features. To further study the ability to extract subtle information in local representations, we performed semantic segmentation with frozen features. As shown in Table 10, SimLAP† has higher mIoU scores except in voc12aug, revealing competitive local characteristics.

**Domain Shift.** Models trained on a dataset may perform poorly when the input distribution changes due to adaptive overfitting. We used several validation sets, different from the original ImageNet dataset, to assess the robustness of the learned representations. For each model, we train a linear classifier using the IN1K training set. We then evaluate the top-1 accuracy on different validation sets to exhibit performance under domain shift. As shown in Table 11, SimLAP† exhibits competitive performance with SL+SSL, which has been proven to solve class collapse (Islam et al., 2021; Xue et al., 2023).

Table 10: **Semantic Segmentation** with frozen features via FCN. We report mIoU for three benchmarks.

| mIoU | SimLAP† | MoCo | GPaCo | SimCLR* | Supcon* |
|---|---|---|---|---|---|
| ade20k | **22.45** | 22.13 | 19.1 | 22.32 | 19.27 |
| cityscapes | **55.77** | 54.48 | 47.41 | 54.49 | 49.69 |
| voc12aug | 48.56 | 52.83 | **56.09** | 44.80 | 39.35 |

Table 11: **Domain shift of IN1K**. We report top-1 accuracy on origin, ImageNet-Sketch (Wang et al., 2019) and ImageNetv2 (Recht et al., 2019) (matched, threshold0.7, and top-images). SL+SSL denotes a combination of SL and MoCo (Islam et al., 2021).

| top-1 | origin | Scketch | matched | threshold0.7 | top-images |
|---|---|---|---|---|---|
| SL | 76.1 | 22.4 | 63.3 | 72.7 | 78.1 |
| SL+SSL | **76.7** | 26.1 | **64.8** | **73.6** | 78.3 |
| SimLAP† | 76.0 | **29.3** | 64.1 | **73.6** | **78.83** |

**Scaling with ViTs.** Vision transformers have an increasing importance on computer vision. To study the compatibility with ViTs, we trained SimLAP with ViTs in different model sizes. Each model was trained for 300 epochs with the AdamW optimizer (blr=5e-4). Table 12 shows the CIFAR10 KNN performance for three ViT variants. While our method has competitive performance, the gains from scaling model size were modest.

Table 12: **Scaling with ViTs**. We report top-1 accuracy on CIFAR10 with KNN.

| | ViT-tiny | ViT-small | ViT-base |
|---|---|---|---|
| Supervised | 76.94 | **92.38** | **93.49** |
| SimLAP | **82.56** | 91.2 | 92.75 |

**Arbitrary Pair vs. Instance Pair.** To compare learning with arbitrary pairs and with instance pairs, we consider two popular CL frameworks (SimCLR and MoCo). The instance-wise CL methods can learn with arbitrary pairs by introducing our feature filter, resulting in SimLAP and MSimLAP. We assess these models on CIFAR10 using KNN accuracy over different training durations. As shown in Figure 11, converting CL with instance pairs to arbitrary pairs consistently improves performance. The performance gain is significant for all scenarios, ranging from 1.4% to 1.8%. Notably, MSim-LAP outperforms MoCo (trained for 1000 epochs) within a significantly shorter training duration (300 epochs). While KNN accuracy on CIFAR-10 provides a limited evaluation scope, the observed trends strongly suggest that cross-class CL is more efficient and effective than traditional instance-wise CL.

## D HYPERPARAMETER

We study the impact of hyperparameters for training. Each experiment was trained on IN1K for 100 epochs. The batch size was 1024. We apply the linear evaluation (Chen et al., 2020) on IN1K and KNN (K=10) on CIFAR10 to assess the learned representations. Table 13 demonstrates the effects of various hyperparameters. 'act.' calculates the average sum of gate vectors for class pairs, indicating the mean dimensionality of subspaces. Default settings for training are highlighted in under line.

**Normalization.** We find that normalization at the end of the projector is critical to train SimLAP. Without normalization, the training process becomes unstable, resulting in NaN errors. Conversely, the model barely converges with Batch Normalization (Ioffe & Szegedy, 2015). This is because Batch Normalization enforces each dimension to be informative (high variance), conflicting with our filter module, which selectively eliminates certain dimensions. Our model only functions effectively with Layer Normalization (Ba et al., 2016).

**Training Parameter.** Similar to other CL methods (Chen et al., 2020; He et al., 2020), SimLAP's performance is influenced by many factors.

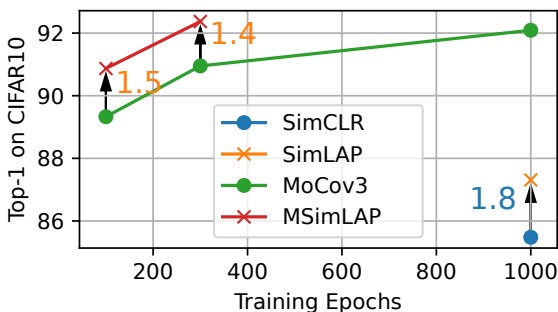

Figure 11: **Converting instance pairs with arbitrary pairs.** We report the KNN accuracy on CIFAR10. MSimLAP denotes SimLAP†.

| Norm | IN1K | CF10 |
|------|------|------|
| LN | 70.5 | 82.9 |
| BN | 2.2 | 18.5 |
| None | NaN | NaN |

(a) **Normalization**.

| opt | IN1K | CF10 | act. |
|-----|------|------|------|
| AdamW | 70.5 | 82.9 | 192 |
| LARS | **70.6** | **86.3** | 83 |

(b) **Optimizer**.

| Dim | IN1K | CF10 | act. |
|-----|------|------|------|
| 256 | 70.5 | 82.9 | 192 |
| 2048 | 70.3 | 82.9 | 306 |
| 4096 | **70.6** | **84.1** | 437 |

(c) **Dimensions** of features.

| $\tau$ | IN1K | CF10 | act. |
|--------|------|------|------|
| 0.05 | **71.2** | **84.11** | 70 |
| 0.1 | 69.8 | 82.9 | 108 |
| 0.15 | 70.5 | 82.9 | 192 |

(d) **Temperature** of CL.

Table 13: **Tuning hyperparameters**. *⎽* denotes the default setting for tuning hyperparameters. We report Top-1 accuracy of the linear prob (IN1K) on IN1K and KNN ($k = 10$) on CIFAR10. Layer Normalization is vital to train SimLAP. LARS, high-dimensional projector, and $\tau = 0.05$ are the suitable for SimLAP.

Table 13b compares optimizers. AdamW (Loshchilov & Hutter, 2019) are more efficient in training the neural networks. However, LARS (You et al., 2017) converges to a better solution by 3.4% accuracy improvement. Optimization affects the average activation.

Table 13c illustrates the effect of varying the projector's dimensions. While a large projection space is crucial for transfer learning, increasing dimensionality has minimal impact on linear probing. This is because 256 dimensions suffice for major visual representations, but minor visual information requires more dimensions.

We examined three temperature values (see Table 13d). While exploring more values might yield further improvements, our chosen value ($\tau = 0.05$) effectively demonstrates cross-class feature learning within resource constraints. Notably, average activation correlates with $\tau$.

## E    BENCHMARK

**IN1P.** To investigate the model's ability to extract shared information among related classes, we created the IN1P dataset. This dataset comprises ten dog breeds selected from ImageNet-1K: Chihuahua, toy terrier, Walker hound, English foxhound, Saluki, Chesapeake Bay retriever, Rottweiler, Doberman, boxer, Great Dane. Figure 12 presents a sample image from each class in the IN1P dataset. Despite the variations in breed, common features characteristic of dogs are evident across all samples. This carefully curated dataset allows us to examine how effectively our model can identify and leverage shared features among closely related, yet distinct, classes.

**IN25.** We introduce IN25, a carefully curated subset of ImageNet-1K designed to study how the semantic distance of positive pairs affects contrastive learning. The dataset comprises 25 classes organized into 5 super-classes: cars, snakes, birds, dogs, and cats, with each super-class containing 5 sub-classes (Figure 13). This hierarchical structure creates multiple levels of semantic relation-

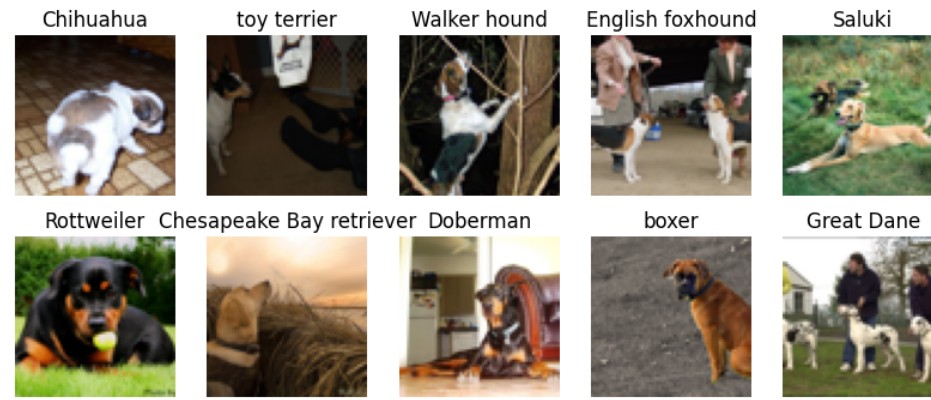

Figure 12: Random samples from IN1P.

ships: Intra-super-class: Classes within the same super-class (e.g., different dog breeds) exhibit high semantic similarity; Inter-super-class: Classes across super-classes demonstrate varying degrees of semantic distance (e.g., dogs are semantically closer to cats than to cars). To quantify these semantic relationships, we analyze class similarities using CLIP (ViT-B/16) embeddings. Figure 14a presents a heatmap of average similarities between all class pairs, revealing clear block-diagonal patterns that correspond to super-class groupings. For a more detailed view, Figure 14b shows similarity distributions relative to a single class (Golden Retriever), demonstrating how semantic distances vary continuously across different super-classes.

**Downstream Tasks.** We use 8 downstream tasks to evaluate the transfer learning performance: **CIFAR-10**: A widely-used dataset for image classification, consisting of 60,000 32x32 color images across 10 classes, with 6,000 images per class. It includes common objects such as airplanes, automobiles, birds, and cats. The dataset is split into 50,000 training images and 10,000 test images. **STL-10**: Inspired by CIFAR-10 but designed with unsupervised feature learning in mind. It contains 5,000 labeled images across 10 classes and 100,000 unlabeled images. The images are larger (96x96) and of higher quality compared to CIFAR, making it more challenging and realistic. **Stanford Cars**: A fine-grained visual classification dataset containing 16,185 images of 196 classes of cars. The data is split into 8,144 training images and 8,041 testing images. **Oxford-IIIT Pet Dataset**: Consists of 7,349 images of cats and dogs across 37 breeds. The dataset features 12 cat breeds and 25 dog breeds, with roughly 200 images per class. It's commonly used for tasks such as fine-grained classification and segmentation. The data is split into 3,680 training images and 3,669 testing images. **FGVC-Aircraft**: A benchmark dataset for the fine-grained visual categorization of aircraft. It contains 10,200 images of aircraft, with 100 images for each of 102 different aircraft model variants, most of which are airplanes. The models are organized in a four-level hierarchy: Model, Variant, Family, and Manufacturer. The data is divided into three equally-sized training, validation, and test subsets. **Describable Textures Dataset (DTD)**: An evolving collection of textural images in the wild, annotated with a series of human-centric attributes, inspired by the perceptual properties of textures. The DTD consists of 5,640 images, organized according to a list of 47 terms (categories) inspired by human perception. There are 120 images for each category, with image sizes ranging between 300x300 and 640x640. The images contain at least 90% of the surface representing the category attribute. The data is split into three equal parts for training, validation, and testing, with 40 images per class for each split. **Food-101**: A dataset for food image classification, consisting of 101 categories, with 101,000 images. Each class includes 250 manually reviewed test images and 750 training images, intentionally left uncleaned to include some noise, which manifests as intense colors and occasional incorrect labels. To standardize the images for processing, all images have been rescaled to have a maximum side length of 512 pixels. **Oxford Flowers-102**: A fine-grained image classification dataset comprising 102 flower categories. It contains 8,189 images in total, with each class consisting of between 40 and 258 images. The training set has only 2,040 samples. The dataset is particularly challenging due to the fine-grained nature of the categories and the large variations in scale, pose, and lighting conditions. These tasks involve general image clas-

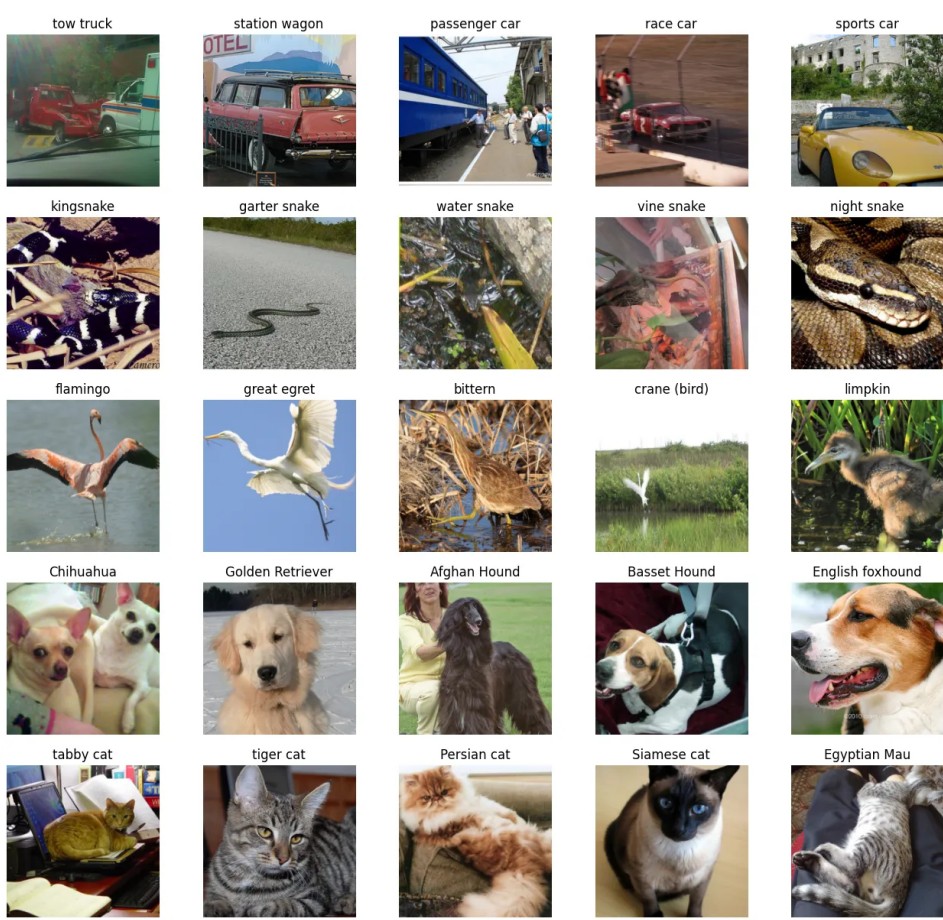

Figure 13: Random samples from IN25.

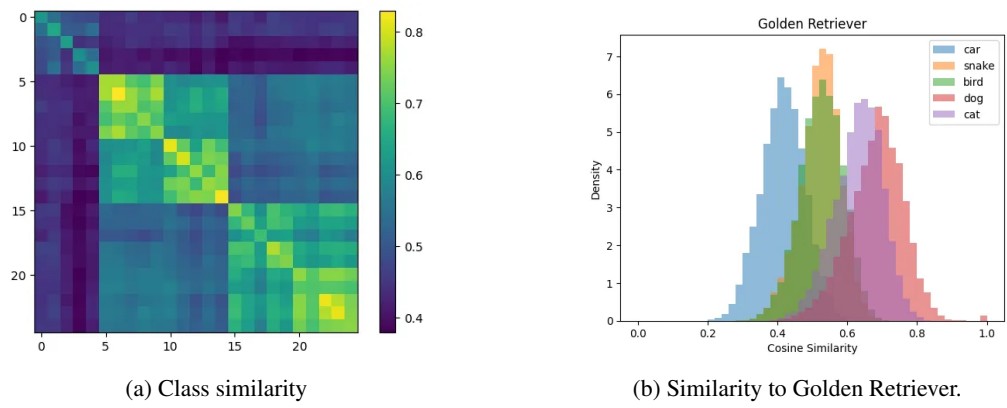

(a) Class similarity

(b) Similarity to Golden Retriever.

Figure 14: IN25 demonstrates hierarchical semantic distances between classes.

sification tasks and fine-grained classification tasks, providing a comprehensive evaluation of the learned representations for the image.

**ROxford5k and RParis6k**. In the revised ROxford5k and RParis6k benchmarks (Radenović et al., 2018), each query-associated positive image is annotated with one of three difficulty labels——Easy, Hard, or Unclear——according to the magnitude of viewpoint, illumination, and occlusion variation relative to the query crop. The Medium evaluation protocol treats all images marked Easy or Hard

Table 14: The Importance of the Gating Mechanism in Breaking Symmetry for Arbitrary Pairs. Only methods employing a Gating mechanism ('Y', SimLAP) successfully resolve the conflict arising from arbitrary pairs, achieving performance comparable to or better than identical-pair methods.

| | Supcon | | AVG | mixup | SimLAP | |
|---|---|---|---|---|---|---|
| Pairs | identical | arbitrary | arbitrary | arbitrary | identical | arbitrary |
| Gate | N | N | N | N | Y | Y |
| Acc | 89.91 | 25.34 | 27.33 | 26.5 | 90.58 | **90.86** |

as positives, whereas the Hard protocol regards only the Hard subset as relevant, ignoring Easy and Unclear instances entirely. Consequently, Medium demands retrieval of both canonical and moderately challenging views, while Hard exclusively rewards localisation under severe appearance changes.

## F    GATING MECHANISM

SimLAP resolves the conflicts arising from pulling distinct pairs by dividing representations into subspaces via gating vectors. The gating mechanism is a vital component to the success of our method. Therefore, we will demonstrate the gating mechanism through a toy example on the CIFAR-10 dataset.

**Setting.** We train a ResNet-18 model on CIFAR-10 for 50 epochs, optimized using the Adam optimizer. The learning rate is set to $0.001$, and the batch size is $256$. Trained models will be evaluated using a linear probe on CIFAR-10, which involves training a linear layer that takes frozen features from the backbone network as input.

**Breaking Symmetry.** Arbitrary pairs introduce a symmetry structure of pulling and pushing classes, which is the root cause of the failure in traditional contrastive learning methods. This symmetry implies that samples from any given class have an equal tendency to reduce the distance to other classes, preventing models from effectively distinguishing between them.

Our gating mechanism is designed to break this symmetry by selectively activating dimensions of the representation. To demonstrate the critical importance of the gating mechanism, we compare our approach against two baseline methods that also reduce distance but lack gating:

1. **AVG**: Calculates similarity using $\langle z_1, (z_1 + z_2)/2 \rangle$.
2. **Mixup**: Calculates similarity using $\langle z_1, f(\bar{x}) \rangle$.

where $z = f(x)$ and $\bar{x} = \lambda x_1 + (1 - \lambda)x_2$, $(x_1, x_2)$ is a random pair.

Although AVG and Mixup both attempt to reduce the distance to the anchor, they simultaneously pull a distinct (or arbitrary) sample, which ultimately leads to a failure to optimize the learned representations effectively. Table 14 clearly demonstrates that only our gating mechanism (SimLAP-Arbitrary) successfully breaks this detrimental symmetry for arbitrary pairs, achieving high accuracy.

We further visualize the class similarity measured by gates, global features, and label vectors in . Our gating mechanism break the symmetry between classes, because the filter could inherit the asymmetric property from visual characteristics via optimizing subspaces. Similar class pairs will have larger size of subspaces, which cause more contribution or gradient to update representations. Therefore, the visual information brings asymmetric visual representations. Then, label embeddings capture this asymmetric structure, resulting asymmetric gate vectors. Eventually, asymmetric gate vectors break the symmetric objective of pulling arbitrary pairs.

## G    UNDERSTANDING FROM A CASE STUDY

We visualize the gate values of SimLAP in Figure 15. Specifically, we pass 1K classes to the filter to generate 1000 vectors with 256-D gate values, which indicate the activation of the corresponding dimensions for each class. Each light point on the heatmap represents the activation status of a

specific dimension for the corresponding class. From the figure, we can observe that the gates exhibit roughly binary values (either activated or not), and the activation vectors for different classes are distinct. This indicates that each class creates a unique subspace within the feature space. These experimental observations support our hypothesis that SimLAP can effectively learn to create class-level subspaces.

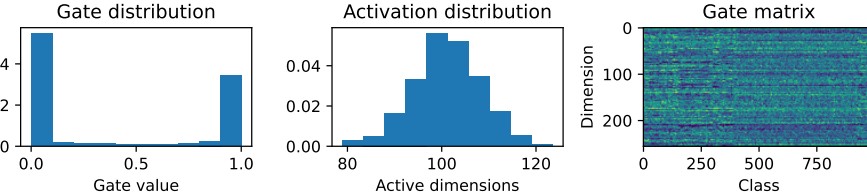

Figure 15: **Gate Visualization**. The activated dimensions are identified by the sum of gate values for each class, defining the size of the subspace. Gate matrix denotes the activation of one dimension for a class. The filter selects different dimensions for each class, demonstrating the creation of class-specific subspaces.

Considering the large number and diversity of classes in IN1K, we use a case study to demonstrate how SimLAP interprets the similarity between classes. Specifically, we select 17 classes belonging to the super synset 'snake.n.01', 16 classes belonging to the super synset 'wading bird.n.01', and 21 classes belonging to the super synset 'furniture.n.01' from the IN1K validation set. Each class contains 50 images. This case study help us to understand that SimLAP promotes cross-class features as well as class-level features.

**Semantic Hierarchy.** Synsets are sets of cognitive synonyms, such as 'hunting_dog.n.1', defined in WordNet (Fellbaum, 1998). Synsets are inter-linked by a sparse network of typed conceptual relations (hypernymy, hyponymy, meronymy, etc.). Each synset carries a concise definition ("gloss") and, wherever useful, one or two illustrative sentences. ImageNet is organised according to WordNet, thus we know which classes in ImageNet belongs to a super synset, e.g., 21 classes belonging to the super synset 'furniture.n.01'.

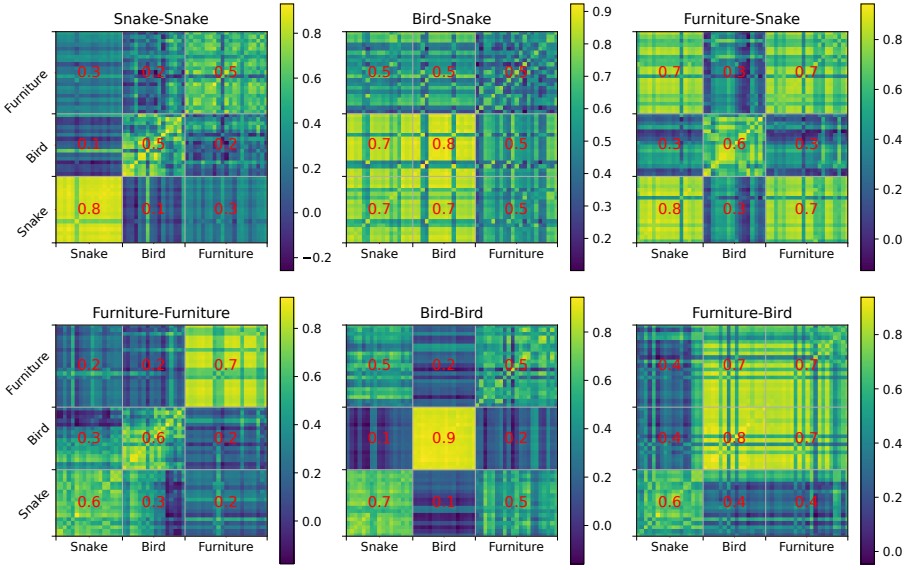

Figure 16: **Class similarity in subspaces.** The gap between intra and inter coarse-class similarity is widened in their corresponding subspaces, highlighting SimLAP's effectiveness in creating discriminative subspaces.

**Subspace of Common Features.** We investigate how SimLAP identifies common features between similar classes within a coarse category. For the subclasses within a coarse class, the common

features are dominant, i.e. showing high similarity in the global space, as discussed earlier. To delve deeper, we select the top 10 dimensions with the highest activation for subclasses within each coarse class, as illustrated in Figure 17. These dimensions likely represent the most salient features for distinguishing the subclasses. We then calculate the class similarity of features in the subspace defined by these dimensions for each coarse class. Figure 16 visualizes the class similarity in the subspaces computed for three coarse classes: Snake, Bird, and Furniture. Compared to the class similarity in the global space, the discriminative information is strengthened for the corresponding subspace. Specifically, the gap between intra and inter coarse-class similarity is magnified to 0.5 for Snake in the Snake-Snake Subspace, 0.7 for Bird in the Bird-Bird Subspace, and 0.5 for Furniture in the Furniture-Furniture Subspace. The features in the subspaces must contain distinct information about the class pair. These features also contain visual information about other classes such that these classes are still discriminative in the unrelated subspaces. These results demonstrate SimLAP's ability to create meaningful subspaces that amplify the differences between the classes, even when no common features appear to exist in the global space, like Furniture-Snake.

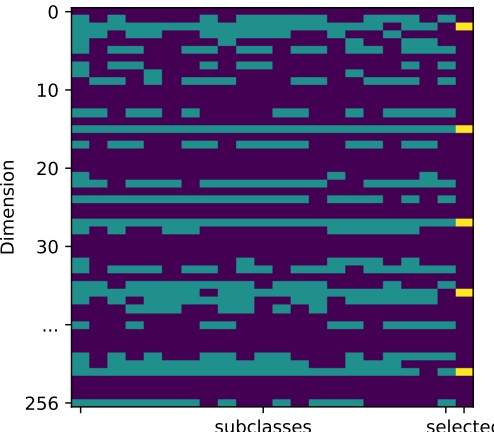

Figure 17: Illustration of **selecting features for garter-lamp**. Dimensions with the highest average activation are selected (yellow).

**Class Similarity in Global Space.** We investigate three similarity measurements to evaluate the ability to identify the relationships between classes: 1) *Dot product of gates*: We calculate the gates for each class and sharpen the values to 1 if above 0.5 and to 0 otherwise. This measure indicates how similar the activated dimensions are between classes, reflecting the structural similarity of their subspaces. 2) *Features in the global space*: For each class pair, we use the global features to calculate the cosine similarity between all sample pairs coming from each class pair. We then use the average to denote the similarity between classes. This represents the overall visual similarity in the original feature space. 3) *Class similarity in label embedding*: We obtain the class vectors through the label embedding in the feature filter, and then calculate their cosine similarity. This measure reflects the learned semantic relationships between class labels. Figure 4 compares these three similarity measurements. All results indicate that SimLAP automatically discovers both semantic and visual similarities between classes, with clearer distinctions between the super synsets compared to within them. This suggests that SimLAP effectively captures the hierarchical structure of the classes, maintaining strong similarities within super synsets while preserving distinctions between them.

## H    LEARNING STABILITY

We investigate the learning stability of arbitrary pairs. We train SimLAP with resnet50 for 100 epochs over 3 seeds and evaluate these models on downstream tasks with KNN (k=10) as shown in Table 15. These intervals demonstrate low dispersion, confirming that the central values are already stable across seeds.

Table 15: **Training SimLAP over three seeds.** We report top-1 accuracy via KNN (k=10).

| seed | DTD | Flowers | Food | Pets | STL | CIFAR10 |
|------|------|---------|------|------|------|---------|
| 1 | 59.6 | 72.3 | 54.6 | 87.3 | 96.5 | 83.7 |
| 2 | 62.8 | 70.8 | 57.3 | 87.7 | 97.0 | 85.6 |
| 3 | 61.1 | 70.0 | 55.9 | 83.0 | 96.2 | 86.8 |

# I  LLM USAGE

We leverage a large language model (LLM)[8] to identify related work by find related work about using common features. Grammatical correctness is enforced within Overleaf via the Writefull plug-in. The proof presented in Appendix A.1 is examined by Gemini. ChatGPT is further consulted to corroborate the novelty of learning from arbitrary pairs; the model cautions that such an approach is viable only when an appropriate inductive bias is imposed, and its proposed solutions diverge materially from SimLAP.

---

[8]https://www.kimi.com/

