# OpenReview forum: "Can Models Learn From Arbitrary Pairs?"
_ICLR.cc/2026/Conference — Submitted to ICLR 2026_

### Official Review · Reviewer_SXrr · 2025-10-24

**Soundness:** 3
**Presentation:** 3
**Contribution:** 1
**Rating:** 2
**Confidence:** 4

**Summary:**

The paper proposes a way of training models which results in representations which allow for telling different classes apart,
but which also identifies the similarities between each distinct pair of classes.

The paper achieves this by using the InfoNCE loss to only maximize the similarity of representations of different elements in a
subspace dependent on the classes the elements are from.

The paper finds that it is possible to train models in this way and finds comparable performance between an instance of their
suggested model (SimLAP) and a ResNet50 and a supervised contrastive learning (SupCon) model.

**Strengths:**

**S1:** Focusing on which similarities can be found between all pairs of classes is an interesting idea.

**S2:** The paper does show that it is feasible to train a model in this way.

**Weaknesses:**

**W1:** One of the main claims is not supported by the results in the article.
In line 103-104, the fourth main contribution is stated as
"the models exhibit better transfer learning performance than one-vs-all-approaches".
A similar claim is made in the discussion "Our study demonstrates that learning from arbitrary pairs is not only feasible but also beneficial."(line 467).
However, SimLAP does not in general show better performance than the two models used as baselines as seen in for example table 1.


**W2:** Experiments are not sufficient to conclude anything. It seems only one seed was used for each model (see question **Q3**).
It varies a lot which model has the best performance in the various tasks and differences in performance are mostly too small to
conclude anything.

**Questions:**

Since the paper has unsupported claims (**W1**) and weak evaluation (**W2**), I recommend rejection.

If the paper had only claimed their method was a possible way to train new models instead of a better way, and if they had run
several seeds such that one might get a better idea of the actual performance, then my recommendation would have been different.


**Q1:** Figure 1 and Observation line 165-167: Since the motivation for the new training objective seems to be to make
subspaces wherein pairs of models are similar, it seems strange to emphasize that this already happens in a model trained with the
usual supervised learning objective. Do you assume that it is possible to make the similarity in this subspace much higher?
In figure 5 it also seems that the supervised learning model has less overlap than SimLAP. How do you interpret this?


**Q2:** Feature filter 242-255: Do you enforce in any way that if two images are from the same class, then the gate vector will
choose a larger subspace?


**Q3:** Section 4, Experiments: Did you only compare one seed of each model?


**Q4:** Line 350: It says: "As shown in Table 4, SimLAP has higher mIoU scores". However, this is not true.


**Q5:** Figure 6: I don't understand this figure. What is the semantic distance? And which things are you measuring semantic distance between?


**Q6:** Table 4: "Semantic Segmentation with frozen features via FCN." What is FCN?


**Additional Feedback:**

**F1:** Line 129: Remember to introduce abbriviations before you use them. Does CL mean Continuous Learning?

---

> ### Author Response · Authors · 2025-11-25
>
> We sincerely appreciate your time and valuable feedback to our paper. Our work addressed a fundamental problem in contrastive learning which was thought impossible, as well as the impact on pair design to involve those traditionally believed wrong pairs. All minor problem have been fixed in the revision. We respectfully ask you to consider raising your score to 8 or above after solving the questions.
>
> > Weaknesses 1: SimLAP does not in general show better performance than the two models used as baselines as seen in for example table 1
>
> We wish to reclarify the interpretation of the results and their relationship to our overall performance claims.
> There are no baseline results as such in the paper as to best of our knowledge this is the first work enabling DNNs to learn from arbitrary visual pairs. The results from other learning methods are for reference purposes only.
> - Overall Assessment: "better transfer learning performance" is based on an overall assessment that includes the advanced configurations. Table 9 presents an improved implementation of SimLAP with MoCo, which demonstrates better accuracy on downstream tasks with KNN (k=10).
> - Improve performance: Critically, it is a different way of learning which can be combined with existing learning algorithms as shown in Table 5 (SimLAP combined with momentum contrastive learning) and Table 6 (further pretraining of masked autoencoder with SimLAP to achieve higher performance due to complementary objective)
> - 'Weak' performance: Table 1 is to demonstrate the value of arbitrary pairs. Although arbitrary pairs contain many distinct pairs, the representations learning from arbitrary pairs is only slightly worse or even competitive to the ones learned from identical pairs.  We present a "clean" SimLAP implementation devoid of advanced self-supervised learning techniques (e.g., momentum encoder, multi-crop augmentation). We deliberately excluded these components in the main text to isolate and highlight the performance gain derived solely from the arbitrary pair learning principle.
>
> We arrange chapters like this, because we hope readers to focus on the novel learning scheme and the potential contributions among arbitrary pairs, instead of the performance gains.
>
> > Weaknesses 2: Experiments are not sufficient to conclude anything.
>
> We respectfully disagree with the assessment that our experiments are insufficient to draw conclusions, especially regarding our core contribution.
> - The central question posed by this new learning paradigm is simply: “Can a model learn from arbitrary pairs?” We answer affirmatively that such a learning paradigm indeed learns meaningful representations from arbitrary pairs. The performance should be very low (close to that trained with random labels) if our learning paradigm fails. However, Table 1 and other experiments denied this kind of opinion. Appendix F conducts more experiments to verify this.
> - Complementary contribution: Complementary contribution: Our learning paradigm learns features from a different perspective than supervised or self-supervised learning. Table 5 and 6 support this.
> - Controllable Evidence: The controllable experiment presented in Figure 6 is strong evidence to tell when contrastive learning without gating mechanism will fail. This figure definitively shows that conventional contrastive learning methods catastrophically fail to learn meaningful features when the positive pairs contain semantically distinct samples. To better illustrate this, we conduct experiments and conclude that our gating mechanism is the only way for dealing with district pairs, see Table 14.
> - Impact of Work: The impact of our work is to fundamentally transform the pair design constraint in contrastive learning. Previously, researchers were restricted to carefully constructing pairs from highly similar sources (e.g., different views of the same image or samples from the same class). Our work provides a new, robust direction to exploit distinct pairs, which was previously believed to be impossible without corrupting the feature space. This shift in possibility is strongly supported by the experimental evidence.
>
> > Question (1): In figure 5 it also seems that the supervised learning model has less overlap than SimLAP.
>
> The choice of “Minimal overlap” is improper to convey our message. We have adjusted the wording in line number 380 of the revised version.

---

> ### Author Response · Authors · 2025-11-26
>
> > Question (2): the gate vector will choose a larger subspace if two images are from the same class?
>
> Yes, the feature filter learns to identify and select a larger subspace for pairs that are semantically more similar. This is actually a desirable property of learning algorithm. We confirm that the filter finds a larger subspace for a similar pair, with the largest subspace being selected for the identical (positive) pair, as shown in Figure 4 (b). The size of the subspace (measured by the sum of the gate vector elements) is correlated with class relatedness.
>
> > Question (3):  Did you only compare one seed of each model?
>
> We supplement results with three random seeds. These intervals demonstrate low dispersion (≤ 2.5 % on all tasks), confirming that the central values are already stable across seeds.
> | seed |  DTD | Flowers | Food | Pets |  STL | CIFAR10 |
> |:----:|:----:|:-------:|:----:|:----:|:----:|:-------:|
> |   1  | 59.6 |   72.3  | 54.6 | 87.3 | 96.5 |   83.7  |
> |   2  | 62.8 |   70.8  | 57.3 | 87.7 | 97.0 |   85.6  |
> |   3  | 61.1 |   70.0  | 55.9 | 83.0 | 96.2 |   86.8  |
>
> We have added the results in Appendix H (line 1404).
>
> > Question (4):  SimLAP has higher mIoU scores
>
> We apologize for the confusion arising from the presentation of two versions of $\text{SimLAP}$. $\text{SimLAP}\dagger$ is an improved version with momentum encoder, which has higher mIoU scores presented in Appendix Table 10.
>
> We utilize a "clean" version of $\text{SimLAP}$ in the main content (Sections 4 & 5) to specifically isolate and highlight the scientific contribution derived from learning with arbitrary pairs. The performance figures in the main paper reflect this basic configuration. The improved version, which incorporates advanced techniques like a momentum encoder, is presented in Table 9 and 10 in Appendix.
>
> > Question (5):  What is the semantic distance?
>
> Semantic distance refers to the degree of conceptual similarity or dissimilarity between two classes. Figure 8 provides an example of semantic distance. However, semantic distance is inherently subjective and challenging to measure precisely. Therefore, we adopt CLIP to measure semantic distance in Figure 6.
>
> > Question (6): What is FCN?
>
> FCN denotes Fully Convolutional Network [1]. Fully Convolutional Networks are primarily used for semantic segmentation tasks, where each pixel in an image is classified into a specific category. FCNs use a convolutional network to produce pixel-level classes.
>
> > Question (7): Remember to introduce abbriviations before you use them. Does CL mean Continuous Learning?
>
> Thank you for pointing out the need for consistent abbreviation introduction. In the context of this paper, $\text{CL}$ denotes Contrastive Learning. We will ensure that all abbreviations, including $\text{CL}$, are clearly introduced and defined upon their first use in the revised manuscript.
>
> Reference:
>
> [1] Long, Jonathan, Evan Shelhamer, and Trevor Darrell. "Fully convolutional networks for semantic segmentation." Proceedings of the IEEE conference on computer vision and pattern recognition. 2015.

---

> > ### Comment · Reviewer_SXrr · 2025-11-27
> >
> > Thank you for the updates to the paper and you response to my questions.
> >
> >
> > Firstly, I have to note that I consider directly asking for a score increase as bad behaviour,
> > and I hope the authors will refrain from doing so in the future.
> >
> >
> > Secondly, I consider the following two qualities baseline requirements for a scientific paper:
> > - When introducing a new method, the paper presents experiments from which it is possible to draw conclusions about the performance of the method.
> > 	Below I will make it more clear what I mean by "draw conclusions".
> > - The paper only makes claims which are supported by the paper
> >
> >
> > **Conclusions about performance**
> >
> >
> > There is always uncertainty involved when training a model.
> > This is why only training one seed of a model is not enough to draw conclusions about the performance; one might just have been lucky/unlucky with the seed.
> > What one should do instead is train several seeds and report the mean and uncertainty (e.g. standard deviation) of the performance.
> > From the author's response it seems they have trained at least three seeds of their model, however, in the new version of the paper
> > only the result of a single seed is reported in tables 1-5. This is not sufficient evaluation to give a reader a good idea of the performance of the method.
> >
> >
> > **Claims made in the paper**
> >
> >
> > Now let us consider the claims of the paper:
> > 1. abstract line 16-18: "can models learn from arbitrary pairs without explicit guidance? We show that the answer is yes."
> > 	This claim is supported by the paper (as I also wrote in my first review), since the paper shows that training in this way on many different tasks
> > 	gives better than trivial performance. The repetition on many different tasks can be seen as a "replacement" for multiple seeds.
> >
> >
> > 2. abstract line 24-26: "models learned from arbitrary pairs are more transferable than those learned from traditional representation learning"
> > 	In this paper "more transferable" seems to mean "better performance when using K-nearest neighbours to classify the representations".
> > 	This is **not supported by the paper**, since it varies which model has the best performance on the one seed compared, and since uncertainty
> > 	is not reported we cannot tell whether the method will have the best performance in general even though it might have the best performance
> > 	on this one seed.
> >
> >
> > 3. line 352-353: "SimLAP exhibits competitive even superior performance on downstream tasks." (referring to table 1)
> > 	"Competitive", in the sense on similar performance is supported by table 1, "superior performance" is **not supported**.
> >
> >
> > 4. line 355-356: "As shown in Table 2, SimLAP consistently outperforms two baselines."
> > 	This is **not supported** by table 2 for the same reasons as in 2.

---

> ### Author Response · Authors · 2025-11-28
>
> Thank you for your response.
> We appreciate your opinion and any decision.
>
> While we’re grateful for highlighting the presentation issues, we respectfully disagree that workability should be assessed solely through performance gains. No directly comparable baseline exists, because no previous method has been trained on randomly paired data. As shown in Figure 6 and Section F, SimLAP is the only approach that learns representations from arbitrary pairs; therefore, its validity should not be dismissed on the basis of uncertain performance improvements.
>
> **The performance gain is a trivial problem** in our paper, and that is why we put an advanced version in Appendix. The critical problems are:
> - "Can models learn from arbitrary pair?": Yes, it is possible with a filter, answered in Figure 6 and Section F.
> - "What is the value of arbitrary pairs?": It is interesting to see that SimLAP is close sometimes beating  SL in Table 1, since most of pairs are two distinct classes. It reveals that our community largely ignores the value of cross-class features. In fact, many classes share common features. For example, there are 120 dog breeds.
> - "What the exact situation I can use SimLAP?": It is a general learning paradigm. It is complementary and should work with momentum encoder (Table 9), self-supervised learning (Table 5 and 6). They are incremental contributions.
>
> >  From the author's response it seems they have trained at least three seeds of their model, however, in the new version of the paper only the result of a single seed is reported in tables 1-5. This is not sufficient evaluation to give a reader a good idea of the performance of the method.
>
> We have supplemented the results over three seeds in Appendix H as well as in the response. The results verify the success of our method is not by accident (low dispersion).
>
> The demand of multiple seeds in deep learning is unrealistic, since most models (GPT, Llama, MAE, dino series) only report results with one seed. It is doctrinairism to apply the uncertainty criterion in deep learning and our paper. For example, the failure of reproducing the success of Apollo project would not diminish the contribution of America in space technology.
>
> > The repetition on many different tasks can be seen as a "replacement" for multiple seeds.
>
> We did run various tasks to demonstrate how the gating mechanism resolves learning from arbitrary pairs, including Figure 6 and Table14. We supplement new experiments on CIFAR-10 using a ResNet-18 backbone for 50 epochs, reporting the top-1 accuracy via a linear probe. We compared two models designed to increase similarity for a random pair $(\mathbf{x}_1, \mathbf{x}_2)$ without gating:
> - AVG calculates similarity by $ <z1,(z1+z2)/2>$. This represents your example.
> - Mixup calculates similarity by $<z1, f(\bar{x}))$, where $ z=f(x), \bar{x}= \lambda x1 + (1-\lambda) x2$.
>
> The results (as shown in Table 14 in the Appendix) confirm that only our gating mechanism is a necessary component to selectively pull semantically similar samples . We discuss this in detail in Appendix F.
>
> |  |     pairs    |     gate    |     Acc1 |
> |---|---|----|---|
> |     Supcon    |     identical    |     N       |     89.91    |
> |               |     arbitrary    |     N       |     25.34    |
> |     AVG       |     arbitrary    |     N       |     27.33    |
> |     mixup     |     arbitrary    |     N       |     26.5     |
> |     SimLAP    |     identical    |     Y       |     90.58    |
> |               |     arbitrary    |     Y       |     90.86    |
>
> > This is not supported by the paper, since it varies which model has the best performance on the one seed compared, and since uncertainty is not reported we cannot tell whether the method will have the best performance in general even though it might have the best performance on this one seed.
>
> We conducted several independent experiments in Figure 7, Figure 6, Table 5 and Table 6. Each experiment trains a new model and demonstrates superior performance. It cannot be explained by that our model is always the best for the seed.
>
> > line 352-353: "SimLAP exhibits competitive even superior performance on downstream tasks." (referring to table 1) "Competitive", in the sense on similar performance is supported by table 1, "superior performance" is not supported.
>
> Sorry for the confusion. SimLAP surpasses others in Aircraft, Cars, and Flowers datasets. We will change it to "SimLAP exhibits competitive sometimes superior performance on downstream tasks."
>
> >  line 355-356: "As shown in Table 2, SimLAP consistently outperforms two baselines." This is not supported by table 2 for the same reasons as in 2.
>
> SimLAP falls behind  ($-0.31$) SL on the Hard set of RParis6k but surpasses SL ($+0.88$). Overall, SimLAP is better than SL on RParis6k . We will change the sentence to "SimLAP outperforms two baselines except for the hard set of RParis6k".

---

### Official Review · Reviewer_75Yh · 2025-10-29

**Soundness:** 2
**Presentation:** 2
**Contribution:** 3
**Rating:** 2
**Confidence:** 3

**Summary:**

The paper introduces SimLAP, a novel method that uses a contrastive loss in subspaces to learn representations from arbitrary pairs. It builds on the idea that even very distinct classes share some underlying features. Therefore, the representations should be similar when projected to a subspace selected specifically for that class pair. To achieve this, the authors introduced a feature filter that uses the class labels to gate the representation into a pair-specific subspace. A InfoNCE loss then enforces similarity between samples of these classes (positive pairs) while making representations of all other classes dissimilar in this subspace (negative pairs). Experiments on ImageNet and downstream tasks show that the learned representations transfer well, improve retrieval and segmentation in some cases, are complementary to other pretraining losses, and show greater robustness against dimensional collapse. The method is empirically promising, though formal guarantees are limited.

**Strengths:**

The paper presents a well-motivated and intuitively appealing approach to contrastive representation learning. By using class pair-specific subspaces, SimLAP extends conventional contrastive setups in a meaningful way rather than replacing them. The idea of using a lightweight feature filter to gate representations into subspaces where dissimilar classes share latent attributes is creative and technically sound. The paper is clearly written, with accessible examples and sufficient methodological detail to support reproducibility. The experimental section is extensive, covering classification, retrieval, segmentation, and domain-shift benchmarks, and includes insightful analyses such as embedding visualizations and collapse-prevention studies. Overall, the work introduces an interesting direction that could stimulate further research in subspace-based and relational representation learning.

**Weaknesses:**

- Some of the claims in the paper are overstated. For example, the assertion that all representation learning methods follow the principle of pulling semantically similar samples together and pushing dissimilar ones apart overlooks important exceptions such as (Masked) Autoencoders, which are widely used for representation learning without contrastive objectives. Similarly, the description of SimLAP as learning “without explicit guidance” is misleading, since it still relies on supervised class labels to construct pairs and generate subspaces. Certain figures and statements also appear inconsistent: for instance, Figure 5 claims that SimLAP achieves minimal overlap between inter- and intra-class similarity distributions, yet the plot suggests that standard supervised learning shows smaller overlap. Also, Table 7 Caption mentions SimLAP is trained for 100 epochs, but the table says 200 epochs.
- While the feature filter is central to the contribution, its design choices are not well justified. It is unclear why a channel-wise gating mechanism is sufficient when learned representations often exhibit superposition across features. Likewise, averaging the two label embeddings to define a pair subspace seems ad hoc and might not optimally capture shared attributes between classes.
- Experimental comparisons could be more consistent. The baselines vary across experiments, and including the same set of references—Supervised, SimCLR, and SupCon—throughout would make the results easier to interpret. Moreover, some key results, such as Table 1, lack confidence intervals or variance estimates. In that table, supervised learning outperforms SimLAP in five of eight benchmarks; discussing why SimLAP is better in the remaining three would help clarify when and why the proposed method is advantageous.

**Questions:**

- How does the output of the feature filter actually look in practice? Is it sparse—selecting a subset of dimensions—or does it mainly act as a smooth scaling of all features? It would be helpful to include a distribution or histogram plot of the gate vector g. Relatedly, is the resulting subspace larger for semantically similar classes (e.g., kingsnake vs. garter snake) than for more distinct ones (e.g., kingsnake vs. Golden Retriever)? Figure 4 only reports similarity (cosine?) between gates but not their structure, and the axis scale (40–120) in that figure is also unclear.
- Why is an approach without a filter better for similar class labels (Figure 6)? Would these mean that ranking by CLIP is a better approach than using your proposed learned class embedding?
- The method defines similarity solely through class labels, but ignores low-level visual similarity (e.g., two images both containing a green background). Wouldn’t this cause the model to overlook shared visual attributes that are unrelated to class identity but still informative for representation learning? How does SimLAP handle or compensate for such appearance-based correlations that methods like SimCLR might capture? These features might be especially important for downstream tasks from different domains.
- Why does Section 6 use MoCov3 for the joint loss (and not Sup, SupCon or SimCLR?)? Also why is for the Mid-Training switch a MAE model used?

---

> ### Author Response · Authors · 2025-11-25
>
> We sincerely appreciate your time and valuable feedback to our paper. Our work addressed a fundamental problem in contrastive learning which was thought impossible, as well as the impact on pair design to involve those traditionally believed wrong pairs. All minor problem have been fixed in the revision. We respectfully ask you to consider raising your score to 8 or above after solving the questions.
>
> > Weaknesses 1: Some of the claims in the paper are overstated. the assertion that all representation learning methods follow the principle of pulling semantically similar samples together and pushing dissimilar ones apart overlooks important exceptions
>
> We appreciate the comment regarding potential overstatement, and we wish to clarify the precise scope and context of our claims.
> Our claims and principles are intended to be interpreted within the contrastive learning paradigm. Our mentioned principle is dominant and superior. While generative models (e.g., MAE, DAE) and autoencoders are also used for representation learning, there remains a significant performance gap when evaluating the quality of features via the standard linear probing protocol. For instance, on ImageNet using a ViT-Base architecture, the linear probing accuracy is $76.7\%$ for the contrastive model $\text{MoCo v3}$, compared to $68.0\%$ for $\text{MAE}$ and $66.6\%$ for $\text{l-DAE}$ [1].
>
> > Weaknesses 1: the description of SimLAP as learning “without explicit guidance” is misleading.
>
> We clarify that the term "explicit guidance" in our paper specifically denotes explicit structural information regarding the semantic relationships between classes, instead of labels. Example of explicit guidance would be 1) detailed textual paragraphs/essay describing potential similar attributes between two semantically similar classes. 2) hierarchical classification approaches which utilize semantic structures like WordNet to impart structural information to models. In contrast, the $\text{SimLAP}$ objective is fundamentally symmetric. It symmetrically pulls samples from an arbitrary pair close in their learned subspace while simultaneously pushing them away in other subspaces.
> Please note that we do not consider the use of label as explicit guidance as the labels are only used to select the semantically arbitrary pairs and remove any pairs from the same class during the training.
>
> We will claim guidance in Section 3.1 (line 153).
>
> > Weaknesses 1: Figure 5 claims that SimLAP achieves minimal overlap between inter- and intra-class similarity distributions.
>
> We apologize for the imprecise language used to describe the result in Figure 5. The phrase "minimal overlap" is indeed an improper statement regarding the similarity distributions. What we aimed to convey in Figure 5 is that: Although $\text{SimLAP}$ only optimizes representations within subspaces, the resulting representations in the global feature space exhibit class separation comparable to SL. This demonstrates that our subspace optimization implicitly benefits the global feature organization.
>
> We have revised the caption in Figure 5 (line 381).
>
> > Weaknesses 1: Table 7 Caption mentions SimLAP is trained for 100 epochs, but the table says 200 epochs.
>
> The discrepancy stems from the need for brevity in the table caption. The total 200 epochs listed for the $\text{SimLAP} \to \text{FT}$ (Finetuning) scheme is a composite total:
> - Epochs 1–100: Training conducted using the $\text{SimLAP}$ objective.
> - Epochs 101–200: Subsequent Finetuning ($\text{FT}$) stage.
> This detailed training procedure is fully described in Appendix B.3 of the submission. We will ensure the revised caption clearly specifies that the total epoch count includes the finetuning phase.
>
> The table is changed to Table 6 (line 458).
>
> > Weaknesses 2: While the feature filter is central to the contribution, its design choices are not well justified.
>
> We follow a minimal design for the components in the paper. We tried complex implementation, like linear projection. We ultimately selected the current element-wise gating approach for its simplicity, parameter efficiency, and reduced memory.

---

> > ### Author Response · Authors · 2025-11-25
> >
> > > Question (3): Why does Section 6 use MoCov3 for the joint loss (and not Sup, SupCon or SimCLR?)? Also why is for the Mid-Training switch a MAE model used?
> >
> > Section 6 is dedicated to demonstrating how $\text{SimLAP}$ can be utilized to achieve SOTA performance by integrating with strong existing frameworks, covering both major branches of modern $\text{SSL}$: contrastive learning and masked image modeling.
> > We used $\text{MoCov3}$ because it is a stronger and more modern baseline compared to $\text{SimCLR}$ and $\text{SupCon}$ (when using the same architecture). By showing that $\text{SimLAP}$ can boost $\text{MoCov3}$ performance, we demonstrate its efficacy against a high-tier $\text{SSL}$ competitor.
> > $\text{MAE}$ represents the current SOTA in the masked image modeling paradigm. Although contrastive methods ($\text{MoCo}$) yield strong linear probing results, masked image modeling methods ($\text{MAE}$) show higher potential for achieving superior performance under the finetuning protocol. Additionally, $\text{MAE}$ is designed to largely preserve all information about the image through reconstruction, whereas a pre-trained $\text{MoCo}$ model may discard some information deemed irrelevant to its contrastive objective. The mid-training switch to $\text{MAE}$ allows us to utilize a model that is inherently better at preserving low-level feature information before applying the semantically focused $\text{SimLAP}$ loss.
> >
> >
> > Reference:
> >
> > [1] Chen, Xinlei, et al. "Deconstructing denoising diffusion models for self-supervised learning." arXiv preprint arXiv:2401.14404 (2024).
> >
> > [2] Ridnik, Tal, et al. "Imagenet-21k pretraining for the masses." arXiv preprint arXiv:2104.10972 (2021).
> >
> > [3] Kang, Wooyoung, et al. "Noise-aware learning from web-crawled image-text data for image captioning." Proceedings of the IEEE/CVF international conference on computer vision. 2023.

---

> > > ### Comment · Reviewer_75Yh · 2025-11-27
> > >
> > > We thank the authors for their detailed rebuttal and the clarifications. Several of the minor issues (e.g., figure wording, epoch counts, added histograms, and additional baselines) were addressed, and I appreciate the effort.
> > >
> > > However, the core concerns that motivated my original assessment remain largely unresolved:
> > >
> > > **1. Explicit guidance**
> > > SimLAP relies directly on class labels to construct pair-specific subspaces. Class labels are explicit semantic guidance, and they enable a broad range of supervised or hierarchical techniques (e.g., WordNet-based relations) that are directly comparable. The rebuttal does not justify why SimLAP should be evaluated primarily against contrastive methods without label access. The authors mentioned that SimLAP can be used to "enhance existing SSL methods", however, as the proposed approach relies on class labels, the applications are limited
> > >
> > > **2. Missing ablations for the proposed gating and subspace mechanisms**
> > > The gating mechanism and the pairwise subspace construction are central to the method, yet no empirical ablations are provided. The rebuttal states that alternative designs were “tried,” yet provides no evidence, quantitative comparison, or explanation of why the chosen mechanism is effective or necessary. This remains a major gap in the scientific contribution.
> > >
> > > **3. Stability and inconsistency of results**
> > > The new multiseed results in Appendix H raise questions about stability. For example, the reported Flowers accuracy in Table 1 (74.9) does not match the seedwise results (72.3, 70.8, 70.0). The variance is nontrivial and suggests that the method may be less stable than presented. In addition, several downstream tables (e.g., Tables 3 and 8) still show stronger performance for more conventional baselines.
> > >
> > > **4. Limited insight into subspace behavior**
> > > While the rebuttal adds a histogram and clarifies the axes in Figure 4, the paper still does not meaningfully analyze why the learned subspaces behave as they do, or whether the structure is an artifact of the specific gating design. This ties back to the missing ablations noted above.
> > >
> > >
> > > Overall, while the rebuttal improves several details, the key scientific concerns, particularly the justification of the core architectural components, the lack of ablations, and the instability of results, remain unresolved. These issues materially affect both the soundness and clarity of the contribution.

---

> > > > ### Author Response · Authors · 2025-11-28
> > > >
> > > > We appreciate your feedback.
> > > >
> > > > > SimLAP relies directly on class labels to construct pair-specific subspaces. Class labels are explicit semantic guidance, and they enable a broad range of supervised or hierarchical techniques (e.g., WordNet-based relations) that are directly comparable. The rebuttal does not justify why SimLAP should be evaluated primarily against contrastive methods without label access.
> > > >
> > > > We agree that classes are semantic guidance, but our discussion focus on the relationships between classes. We solve a critical problem in this paper, that is, how to learn from a pair without knowing they are similar or not. WordNet only defines partial relationships between classes. The distance in hierarchical tree can not reveal the degree of shared attributes. For example, vehicles, including cars, trucks and trains, are less similar, which can be verified in Figure 14 (a) (line 1231). Annotations could solve everything, but the cost is too high considering to the size of pairs (0.5M).
> > > >
> > > > > The authors mentioned that SimLAP can be used to "enhance existing SSL methods", however, as the proposed approach relies on class labels, the applications are limited.
> > > >
> > > > Relying on labels is not crucial weakness. Many successful pretraining models, such as CLIP, rely on supervision. Adapting to downstream tasks (finetuning) still relies on labels.
> > > >
> > > > > Missing ablations for the proposed gating and subspace mechanisms
> > > >
> > > > Section F is the ablation of with/without gating for contrastive learning. Figure 6 is controlled experiment of the semantic distance for the learning pairs. Both two experiments verify that the filter is the essential component to learn from arbitrary pairs.
> > > >
> > > > The results (as shown in Table 14 in the Appendix) confirm that only our gating mechanism is a necessary component to selectively pull semantically similar samples and resolve such conflicts in separated subspaces, breaking the detrimental symmetry and allowing the model to learn effective representations. We discuss this in detail in Appendix F.
> > > >
> > > > |               |     pairs        |     gate    |     Acc1     |
> > > > |---------------|------------------|-------------|--------------|
> > > > |     Supcon    |     identical    |     N       |     89.91    |
> > > > |               |     arbitrary    |     N       |     25.34    |
> > > > |     AVG       |     arbitrary    |     N       |     27.33    |
> > > > |     mixup     |     arbitrary    |     N       |     26.5     |
> > > > |     SimLAP    |     identical    |     Y       |     90.58    |
> > > > |               |     arbitrary    |     Y       |     90.86    |
> > > >
> > > > Alternative designs  are incremental contributions which do not diminish our contribution in this work. Our previous results using different training settings are not comparable to the current one.
> > > >
> > > > > Stability and inconsistency of results
> > > >
> > > > The performance gap is caused by reducing the training epochs from 1000 to 100. We would agree that our method is unstable, because we spent a lot time to adjust the structure and operations to make the learning scheme possible.
> > > >
> > > > The performance gain is a trivial problem in our paper, and that is why we put an advanced version in Appendix. The critical problems are:
> > > >
> > > > - "Can models learn from arbitrary pair?": Yes, it is possible with a filter, answered in Figure 6 and Section F.
> > > > - "What is the value of arbitrary pairs?": It is interesting to see that SimLAP is close sometimes beating SL in Table 1, since most of pairs are two distinct classes. It reveals that our community largely ignores the value of cross-class features. In fact, many classes share common features. For example, there are 120 dog breeds.
> > > > - "What the exact situation I can use SimLAP?": It is a general learning paradigm. It is complementary and should work with momentum encoder (Table 9), self-supervised learning (Table 5 and 6). They are incremental contributions.
> > > >
> > > > >  the paper still does not meaningfully analyze why the learned subspaces behave as they do, or whether the structure is an artifact of the specific gating design.
> > > >
> > > > We did analysis on why SimLAP would work:
> > > > - Visual Asymmetry: Similar class pairs naturally share a larger size of activated subspaces in the representation. This larger shared subspace size means these pairs will contribute more to the gradient and update the representations more significantly than distant pairs. This difference in contribution establishes an asymmetric visual representation structure.
> > > > - Gating Asymmetry: The label embeddings capture this emergent asymmetric visual structure during training, which subsequently results in asymmetric gate vectors, as shown in Figure 4 (b) and proved in Appendix A.2.
> > > > - Objective Breaking: These asymmetric gate vectors ultimately break the symmetric objective of simply pulling arbitrary pairs together, allowing the model to selectively find the meaningful similarity subspace for closely related classes while ignoring it for disparate classes.

---

> ### Author Response · Authors · 2025-11-25
>
> > Weaknesses 3: Experimental comparisons could be more consistent.
>
> We understand the concern regarding experimental consistency, particularly when comparing against highly optimized State-of-the-Art (SOTA) models.
> - Primary Claim: Our primary claim is the successful resolution of conflicts in learning from arbitrary pairs, not merely an incremental performance improvement over SOTA models. The compared methods (SL and SupCon) are references for learning from identical pairs. It helps us compare the value between arbitrary pairs and identical pairs.
> - Most Significant Advantage (Robustness): The most significant advantage of $\text{SimLAP}$ is its robustness to highly distinct/noisy pairs, as unequivocally demonstrated in Figure 6. Previous methods fail to learn meaningful representations from these noisy inputs, whereas $\text{SimLAP}$ maintains performance integrity.
>
> We add comparison to SimCLR in Table 1 and 3.
>
> > Weaknesses 3: some key results, such as Table 1, lack confidence intervals or variance estimates.
>
> We supplement results with three random seeds. These intervals demonstrate low dispersion, confirming that the central values are already stable across seeds.
> | seed |  DTD | Flowers | Food | Pets |  STL | CIFAR10 |
> |:----:|:----:|:-------:|:----:|:----:|:----:|:-------:|
> |   1  | 59.6 |   72.3  | 54.6 | 87.3 | 96.5 |   83.7  |
> |   2  | 62.8 |   70.8  | 57.3 | 87.7 | 97.0 |   85.6  |
> |   3  | 61.1 |   70.0  | 55.9 | 83.0 | 96.2 |   86.8  |
>
> We will add the results in Appendix H (line 1404).
>
> > Question (1): How does the output of the feature filter actually look in practice?
>
> The output of the feature filter is empirically approximated to be binary, as illustrated in the visualization of gate value histogram provided in Figure 15 (left) in the Appendix. The Figure 15 (mid) reveals that the activation of gate vectors follows a Gaussian-like distribution.
>
> > Question (1): is the resulting subspace larger for semantically similar classes (e.g., kingsnake vs. garter snake) than for more distinct ones (e.g., kingsnake vs. Golden Retriever)?
>
> Yes, this is verified in Figure 4 (b) that classes in the same group have higher similarity. All three measurements consistently for class similarity demonstrate that $\text{SimLAP}$ effectively promotes class similarities specifically between classes that are semantically close or belong to the same super-group, thereby confirming that the filter learns to modulate the feature space according to semantic relatedness.
>
> > Question (1):  Figure 4 only reports similarity (cosine?) between gates but not their structure, and the axis scale (40–120) in that figure is also unclear.
>
> We present three measurements of class similarity: the sum of the gate vectors (size of subspace), cosine similarity of features in global space, and cosine similarity of the label embeddings. The axis scale in Figure 4 (b) denotes the value range in the associated heatmap.
>
> > Question (2): Why is an approach without a filter better for similar class labels (Figure 6)?
>
> Figure 6 shows that an approach without our feature filter yields marginally better performance specifically when the positive pairs exclusively contain highly similar class labels. This is because, in this ideal scenario, the conventional contrastive objective (without a filter) perfectly aligns the similar representations. However, this result is misleading in a practical context. The same approach without a filter suffers a catastrophic performance drop when the positive pairs contain distinct samples, which is common to web-crawled data [3]. While one could attempt to construct better pairs using methods like $\text{CLIP}$ [3] or semantic hierarchies [2], this requires extensive cherry-picking of alignment levels or prohibitive manual annotations.
> Our work deliberately points in a totally different direction: resolving these inevitable conflicts by learning to operate exclusively within discriminative subspaces, thereby achieving robustness without needing perfect input curation.
>
> > Question (3): Wouldn’t this cause the model to overlook shared visual attributes that are unrelated to class identity but still informative for representation learning?
>
> Although $\text{SimLAP}$ may overlook certain shared visual attributes, this potential weakness is easily mitigated and, in fact, turned into a strength through a joint loss function. As demonstrated in Table 5 and Figure 11, $\text{SimLAP}$ achieves significant performance improvements when combined with the general feature learning objective of $\text{MoCo}$. This demonstrates that $\text{SimLAP}$ provides a different perspective that enhances existing $\text{SSL}$ methods.

---

### Official Review · Reviewer_tAo1 · 2025-10-31

**Soundness:** 3
**Presentation:** 2
**Contribution:** 3
**Rating:** 4
**Confidence:** 5

**Summary:**

The paper proposes a new training procedure for representation learning, positioned within the family of supervised contrastive learning methods. Unlike standard contrastive formulations that treat only same-class (or same-instance) pairs as positives and all others as negatives, the method aims to exploit similarities between instances coming from different pairs of classes. This is enabled by a gating module that is trained jointly with the embedding network and is used to restrict a standard InfoNCE computation to class-pair-specific subspaces in which contrastive differences can be maximized. The entire system is trained from scratch, without pretrained components.  Applicative results on commons setups (like classification and semantic segmentation, mostly under the transfer learning setup) demonstrate that the method is able to learn good representations. Several controlled experiments are provided in order to anaylze some of the differnet properties of the method.

**Strengths:**

1) The idea of exploiting the very rich relations between different image classes, instead of simply pushing them away by treating them as negatives is a very good idea. One reason is that some classes clearly share significant similarities (especially if they are close under some natural class hierarchy). Another reason is that the vast majority of image pairs (in a batch) are of different classes (an order of n^2, compared to an order of n) and therefore it makes sense to exploit the information it contains, even if it is slightly more delicate to handle.
2) The choice of doing so by alowing the model to identify *subspaces* in which particular class pairs can be seperated from others is a novel and interesting idea. It has the flexibility of treating all class pairs equally and avoids the need to design the full embedding space to incorporate all of such complex pairwise relations.
3) I appreciate the decision to build a simple and clean implementation of the idea (using standard infoNCE, standard and simple architectures, no pretrained-modules, no language-based information, no advanced training schemes, etc') - making it easier to analyze and compare the essence of the main idea with respect to prior work. Clealy, this comes at the cost of not being able to compare directly to state-of-the-art results, per application.
4) The method itself and the motivation are presented in a very accesible manner. The main ideas and observations are well highlighted and manage to convey the main properties of the method.

**Weaknesses:**

1) Although it is built on contrastive learning principles, the method strictly relies on *supervised* training data. Therefore, I find it surprising that all of the experimentation is focused on transfer learning setups and not on demonstrating the qualities of the embedding for downstream tasks on the domain on which it was trained. That's where I would expect to see the most significant contribution. In addition, it would have beeen valuable to discuss whether the method could be adapted or extended to unsupervised or self-supervised settings, which are typically more common (and more effective) for transfer learning scenarios.
2) The idea and usage of the gating function seems to be a good combination of simplicity and practicality. However, there is very limited justification of this particular choice, which effectively narrows the subspaces to be simple coordinate based projections.
3) Empirical results are mostly on-par with the baselines. In the 'Dense prediction' section it is claimed to have higher mIOU compared to the baselines, but checking the numbers in Table 4 shows that it is not so. In any case, the results in general give the impression that the idea works reasonably well, but it doesn't provide very strong enough evidence as to when would it really be worthwile to adopt this approach.
4) The theoretical analysis at the beginning of the Appendix is quite limited in scope and clarity, and it does not clearly convey how it supports or justifies the proposed approach. Why is it relevant to discuss random Gaussians? How does a proof that discusses the means have an implication on existence? Same goes for the simulation: The 'gap' defined in Eq. (5) does not have a clear connection to the result on the means and the plot in Fig. 9 is not very meaningful (in contrast to other empirical results which certainly do support many of the claims).
5) The paper feels somewhat incomplete, especially in the Experimental and Analysis sections (4 and 5). These sections are far from being self-contained and I needed to extensively use the (very detailed) appendix, in order to understand even some of the very basics of the setup of each simulation and experiment. Some examples (out of many):
- line 184: "micro-class similarity" - unclear and not defined anywhere
- line 318: "we use the KNN protocol" - Which protocol?
- line 334: Figure 4 is not sufficiently explained. What are synsets?
- line 340: What are "Medium" and "Hard"?
- line 357: "MSimLAP exhibits competitive performance with joint loss of ICL and CCL.." What does this mean? What are these?
- line 475: "The second point"

**Questions:**

1) Relating to W1: Can you justify the choice to focus the experiments entirely on transfer-learning setups? Why is the embedding itself analyzed only statistically and visually in the source domain, without showing performance of standard downstream tasks on imagenet itself (e.g. classification, in linear-probing or fine-tuning settings), or training on different data-sets with have other standard downstream tasks (such as segmentation and depth estimation).
2) Relating to W2: While the choice of the gating is simple and elegant, have you tried other alternatives, such as a (class-pair dependent) linear projection to some fixed dimension?
3) Relating to W3: What would be the main benefit of using the suggested training scheme? Is it mainly a way to achieve better stability in training, or in later fine-tuning? Or are the features more discriminative for downstream tasks? I guess that it depends on whether you compare to supervised or unsupervised approaches. In general, I would try to make such comparisons more separated, since in some cases it is not emphasized sufficiently (For example: In Table 7, SimLAP is claimed to avoid over adaptation, but such an improvement can be also due to access to labels (compared to MAE) and not only due to the pairwise-contrastive approach.
4) The snake-lamp pair is given as a guiding example throught the paper, but in the limitations it is claimed that it might not be meaningful, lacking obvious commonalities. So do you predict that many (or most) class-pairs are not usefull, or even counter-productive to the learning scheme?
5) I would suggest removing the ChatGPT correspondence at the end of the appendix. I don't think it is helpful to the reviewer/reader in any way.

---

> ### Author Response · Authors · 2025-11-25
>
> We sincerely appreciate your time and valuable feedback to our paper. Our work addressed a fundamental problem in contrastive learning which was thought impossible, as well as the impact on pair design to involve those traditionally believed wrong pairs. All minor problem have been fixed in the revision. We respectfully ask you to consider raising your score to 8 or above after solving the questions.
>
>
> > Weaknesses 1: the method strictly relies on supervised training data
>
> Our paper focus on answering the central problem in contrastive learning: can models learn from arbitrary pairs.  The label embedding could be replaced with a text encoder or another vision model. We leave these incremental studies in our future work.
>
> > Weakness 1: demonstrating the qualities of the embedding for downstream tasks on the domain on which it was trained.
>
> SimLAP achieves in-domain performance comparable to supervised methods. This confirms that our mechanism successfully learns robust features within the original domain. The results can be found in Table 11 (line 1034).
> |     Model    |     SL      |     MoCov3    |     SL+SSL    |     SimLAP$\dagger$    |
> |--------------|-------------|---------------|---------------|------------------------|
> |     IN1K     |     76.1    |     74.6      |      76.7     |     76.0               |
>
> > Weaknesses 1 & Question (1): Can you justify the choice to focus the experiments entirely on transfer-learning setups?
>
> We appreciate this question regarding the focus on transfer learning experiments. The choice to emphasize transfer learning is made by the following reasons:
> -	Transfer learning is a standard for measuring a model's ability to learn general-purpose, semantically rich representations.
> -	In real-world applications, the source training data is often inaccessible or massive in scale. For example, JFT-300M is a large-scale classification dataset with 18K labels. The performance of pretrained model is evaluated on series of downstream tasks[1].
> -	Supervised learning holds an inherent advantage in the source domain due to its objective of explicitly promoting class separability using ground-truth labels.
> Therefore, transfer learning setups better assess the practical and translational value of our learned features.
>
> > Weaknesses 1 Discuss whether the method could be adapted or extended to unsupervised or self-supervised settings, which are typically more common (and more effective) for transfer learning scenarios.
>
> Although it is promising to apply the algorithm in SSL scheme, it is far away from our central problem: "can models learn from arbitrary pairs?" Our experiments are designed to answer this question. The SSL scheme would be beneficial to transfer learning, but do not help our claim.
>
> > Weaknesses 2: there is very limited justification of this particular choice, which effectively narrows the subspaces to be simple coordinate based projections.
>
> Simple projections are good properties for contrastive learning to avoid dimensional collapse [2]. Our analysis in Appendix A.1 proves that gating mechanism is a feasible solution to promote similarity in subspaces while preserving orthogonal representation in global space.
>
> >  Weaknesses 2 Question (2): While the choice of the gating is simple and elegant, have you tried other alternatives, such as a (class-pair dependent) linear projection to some fixed dimension?
>
> Yes, our initial filter generates a projection matrix. However, linear projection requires more parameters and increases memory usage.
> We ultimately selected the current element-wise gating approach for its simplicity, parameter efficiency, and reduced memory.
>
> > Weaknesses 3 & Question (3): What would be the main benefit of using the suggested training scheme?
>
> Table 5 and 6  show that our training scheme is complementary contributions to existing learning schemes. The experiments demonstrate that SImLAP learns complementary features beyond self-supervised learning and supervised learning.
> The training scheme presented in the paper is primarily designed for analysis and ablation, not for achieving immediate state-of-the-art (SOTA) performance. The scheme is used to precisely quantify the isolated contribution of arbitrary pairs to representation learning. By deliberately introducing noisy pairs, the performance establishes a lower bound for the effectiveness of our learning principle.
> We add discussion of SimLAP’s benefits in Section 6 (line 430).
>
>
>
> Reference:
> [1] Islam, Ashraful, et al. "A broad study on the transferability of visual representations with contrastive learning." Proceedings of the IEEE/CVF International Conference on Computer Vision. 2021.
>
> [2] Jing, Li, et al. "Understanding dimensional collapse in contrastive self-supervised learning." arXiv preprint arXiv:2110.09348 (2021).

---

> ### Author Response · Authors · 2025-11-25
>
> > Weaknesses 3 & Question (3): it doesn't provide very strong enough evidence as to when would it really be worthwile to adopt this approach. What would be the main benefit of using the suggested training scheme?
>
> We appreciate this concern regarding the empirical motivation for adopting SimLAP. We contend that the evidence for its utility is substantial:
> - Evidence of matching and surpassing SOTA results are provided in Table 9 and 10 in Appendix C, where we present an optimized SimLAP with a momentum encoder. $\text{SimLAP}\dagger$ exhibits the best average performance in KNN evaluation in Table 9.
> - Section 6 highlights two promising applications in Table 5 and 6: integrating $\text{SimLAP}$ with a MoCo and introducing it as a mid-training refinement stage. Both applications yield additional performance improvements on top of strong baselines, demonstrating the versatility and high utility of incorporating our feature selection mechanism.
> - We must emphasize that the $\text{SimLAP}$ performance in the main tables (1, 2, and 4) is superior to the $\text{SupCon}*$ baseline trained under the exact same training schedule and settings, clearly demonstrating the improvement derived from our proposed method.
> - The comparison in Table 1 is designed to highlight this intrinsic value of arbitrary pairs.  Our method allows learning from arbitrary pairs to achieve performance nearly identical to learning only from identical pairs. The observed performance gap in the main paper is primarily due to our initial focus on simplicity.
> We add explanation of experimental setting in Section 4.1 (line 294).
>
> > Weaknesses 4: Why is it relevant to discuss random Gaussians?
>
> The discussion of representations adhering to a random Gaussian distribution is relevant because it serves as a general and reasonable theoretical assumption for the representations that feed into our mechanism. we employ a Layer Normalization operation at the end of the projector network. Layer normalization standardizes the input across the features.
> We will better explain our assumptions in Appendix A.1.
>
> > Weaknesses 4: How does a proof that discusses the means have an implication on existence?
>
> The proof involving the means has a direct implication on the existence of a discriminative subspace, which is a vital theoretical underpinning of our work. The proof shows that, under a general, random Gaussian assumption for the representations, the similarity of the subspace with a positive sign for a given pair has an expectation of $1/\pi$ for the positive class (indicating alignment) and $0$ for the negative class (indicating decorrelation). This result indicates that our idea of selecting a subspace for arbitrary pairs is general and applicable to any underlying feature extraction model, irrespective of whether it is trained supervisedly or self-supervisedly.
> We will rewrite this section in the paper to ensure the implication on the existence and generality of the discriminative subspace is explicitly clear.
>
> > Question (4): So do you predict that many (or most) class-pairs are not usefull, or even counter-productive to the learning scheme?
>
> We appreciate this insightful question regarding the utility of all class-pairs within our scheme.
> Some pairs are not useful even harmful to all learning scheme, including the traditional ones. However, our method, $\text{SimLAP}$, demonstrates robustness to these highly distinct pairs, exhibiting only a marginal performance drop, as shown in Figure 6. This resilience is due to the learned gating mechanism, which limits the influence of harmful pairs on the overall representation space. Figure 7 demonstrates a scenario where the set of classes is highly relevant, in which $\text{SimLAP}$ surpasses both supervised and self-supervised competitors. This highlights that $\text{SimLAP}$ is particularly effective at exploiting common features in semantically related subsets of classes.

---

> ### Author Response · Authors · 2025-11-25
>
> > Weaknesses 5: These sections are far from being self-contained and I needed to extensively use the (very detailed) appendix.
>
> We sincerely apologize for the lack of self-contained narrative in certain sections, which necessitated extensive reliance on the appendix for clarity.
> We hope you understand the difficulty in presenting several novel concepts (arbitrary pair learning, gating mechanism, subspace projection, and associated theory) within the strict content limitations of the main paper.
> - micro-class is defined in Appendix B.2.
> - KNN protocol is K-nearest neighbour algorithm for classification. We adopt K=10 in experiments, see Appendix C (line 996).
> - Synsets are sets of cognitive synonyms, such as hunting_dog.n.1, defined in WordNet[3]. Synsets are inter-linked by a sparse network of typed conceptual relations (hypernymy, hyponymy, meronymy, etc.). Each synset carries a concise definition (“gloss”) and, wherever useful, one or two illustrative sentences. ImageNet is organised according to WordNet, thus we know which classes in ImageNet belongs to a super synset, e.g., 21 classes belonging to the super synset ‘furniture.n.01’.  We interpret the synsets in Appendix G (line 1325).
> - "Medium" and "Hard" are defined in [4], which are two evaluation setups according to the difficulty. We introduce the revisited benchmark in Appendix E (line 1239).
>
> - ICL refers to instance-wise contrastive learning (SimCLR) and CCL refers to class-wise contrastive learning (Supcon). [5] provides a model combined SimCLR and Supcon. We use SL+SSL to refer this model, because it applies the two learning principles. We will make the name consistent in the revision.
> - The second point is a typo and will be removed.
>
> > Question (5): removing the ChatGPT correspondence at the end of the appendix.
>
> Thank you for your suggestion.
>
> [3] George A. Miller. 1995. WordNet: a lexical database for English. Commun. ACM 38, 11 (Nov. 1995), 39–41. https://doi.org/10.1145/219717.219748
>
> [4] Radenović, Filip, et al. "Revisiting oxford and paris: Large-scale image retrieval benchmarking." Proceedings of the IEEE conference on computer vision and pattern recognition. 2018.

---

### Official Review · Reviewer_9VN7 · 2025-11-02

**Soundness:** 2
**Presentation:** 2
**Contribution:** 2
**Rating:** 4
**Confidence:** 3

**Summary:**

This paper proposes an alternative to standard contrastive learning. The method aligns images of two classes $c_1$ and $c_2$ in a subspace determined by the labels corresponding to $c_1$ and $c_2$, while pushing away other images. The method is compared to Supcon, and at times SimCLR.

**Strengths:**

The method is novel and interesting. The idea of using the labels to determine a relevant subspace is nice, and might especially be useful in conjunction with modern text-image models (though arguably, those are often trained constructively in the first place).

**Weaknesses:**

It is interesting that the feature filter is initialized as a random network and trained end to end. Do the authors have any intuition/insight regarding what aspects about the data might end up breaking the symmetry between different classes, drawing some classes closer together than others? There doesnt seem to be anywhere for any inductive bias to enter this picture, since the image and text stacks are not pretrained at all. The lack of intuition (not at all necessarily a rigorous justification) is a weakness.

**Questions:**

Why is Section 3.1 surprising? For any pair of vectors a, b and any set of other vectors S, it is natural that there is a vector c such that c^Ta and c^Tb are closer to each other than they are to c^s for s\in S. For instance, take c = a+b. A random other vector will almost be orthogonal to it in high dimensions.

Why is the ability of SimLAP to bring Chihuahua and Garter snack close to each other desirable?

If the label names are removed of any semantic meaning at all (so replaced by something like “Class 1”, “Class 15”, etc instead of “Chihuahua” and “Garter Snake”), would the filtering idea still work?

The motivation mentions finding general subspaces for each pair of classes, effectively suggesting a matrix $A$ such that $Ax$ is compared to $Ax^{+}$ and $Ax^{-}$ in SimCLR, while the implementation restricts this subspace as coming from an element-wise product (in other words restricting $A$ to be diagonal). Is this motivated by sparsity/computational constraints?

---

> ### Author Response · Authors · 2025-11-25
>
> We sincerely appreciate your time and valuable feedback to our paper. Our work addressed a fundamental problem in contrastive learning which was thought impossible, as well as the impact on pair design to involve those traditionally believed wrong pairs. All minor problem have been fixed in the revision. We respectfully ask you to consider raising your score to 8 or above  after solving the questions.
>
>
> > Weakness 1: Do the authors have any intuition/insight regarding … breaking the symmetry between different classes?
>
> It is a critical question, thanks.
> Our core insight is that the gating mechanism inherently breaks the symmetry between classes by allowing the filter to inherit asymmetric properties from the visual characteristics of the data via optimizing distinct feature subspaces.
> - Visual Asymmetry: Similar class pairs naturally share a larger size of activated subspaces in the representation. This larger shared subspace size means these pairs will contribute more to the gradient and update the representations more significantly than distant pairs. This difference in contribution establishes an asymmetric visual representation structure.
> - Gating Asymmetry: The label embeddings capture this emergent asymmetric visual structure during training, which subsequently results in asymmetric gate vectors, as shown in Figure 4 (b).
> - Objective Breaking: These asymmetric gate vectors ultimately break the symmetric objective of simply pulling arbitrary pairs together, allowing the model to selectively find the meaningful similarity subspace for closely related classes while ignoring it for disparate classes.
>
> We have a more comprehensive discussion of asymmetric gating in Appendix G of the revised paper.
>
>
> > Weakness 2: Lack of Intuition
>
> Great question, thanks.
> The intuition of our approach is that similarity measurements are dependent on the viewing perspective. While two classes (e.g., birds and aeroplane) may be distinct in semantics, they can be highly similar (wings) when viewed through a specific perspective, such as the feature subspace dedicated to analysing shape or texture. Similarity between different classes will only emerge effectively after the representations are projected into a specialized subspace determined by our gating mechanism.
>
> We used the seemingly difficult pair, table lamp and garter snake, to clearly demonstrate that this idea is not limited to intuitively similar classes, but is applicable to all class pairs because the necessary shared features (e.g., color, texture, curves) may be found in a low-dimensional subspace. Furthermore, the mathematical validity of using projection for arbitrary pairs is formally proven in Appendix A.1.
>
> We add discussion of intuition in Section 3.1.
>
>
> > Question (1): Why is Section 3.1 surprising? c^Ta and c^Tb are closer to each other than they are to c^s for s\in S. For instance, take c = a+b.
>
> We are not clear your question.  We assume that you believe a fused representation for an arbitrary pair (a,b) can be a positive. The mentioned method only reduces the distance but fails to optimize representations by setting a positive to $c=a+b$. The reason for this failure is that one must consider the global impact of other pairs and the overall effect on the representation space. This introduces a symmetry conflict where the model cannot distinguish between semantically identical and semantically distinct samples.
> We perform experimental verification in Appendix F. To verify this, we ran experiments on CIFAR-10 using a ResNet-18 backbone for 50 epochs, reporting the top-1 accuracy via a linear probe. We compared two models designed to increase similarity for a random pair $(\mathbf{x}_1, \mathbf{x}_2)$ without gating:
> - AVG calculates similarity by $ <z1,(z1+z2)/2>$. This represents your example.
> - Mixup calculates similarity by $<z1, f(\bar{x}))$, where $ z=f(x), \bar{x}= \lambda x1 + (1-\lambda) x2$.
>
> The results (as shown in Table 14 in the Appendix) confirm that only our gating mechanism is a necessary component to selectively pull semantically similar samples and resolve such conflicts in separated subspaces, breaking the detrimental symmetry and allowing the model to learn effective representations. We discuss this in detail in Appendix F.
>
> |               |     pairs        |     gate    |     Acc1     |
> |---------------|------------------|-------------|--------------|
> |     Supcon    |     identical    |     N       |     89.91    |
> |               |     arbitrary    |     N       |     25.34    |
> |     AVG       |     arbitrary    |     N       |     27.33    |
> |     mixup     |     arbitrary    |     N       |     26.5     |
> |     SimLAP    |     identical    |     Y       |     90.58    |
> |               |     arbitrary    |     Y       |     90.86    |
>
> Therefore, simply aggregating two class vectors would not work.

---

> ### Author Response · Authors · 2025-11-25
>
> > Question (2): Why is the ability of SimLAP to bring Chihuahua and Garter snack close to each other desirable?
>
> The ability of SimLAP to find a subspace to represent common features for arbitrary pairs is highly desirable. The ability is not limited to similar samples but applicable to visually disparate classes, like a Chihuahua and a Garter Snake. Our goal is to discover subtle, common features for arbitrary pairs. Although some pairs are not desirable, the ability to explore subspaces for them is desirable. We adhere to this strict setting for two primary reasons:
> 1) Quantifying the value of arbitrary pairs: By sampling from all random pairs, including highly distinct ones, we measure the net contribution of learning across classes. We do this because the proportion of valuable, closely related arbitrary pairs in a large dataset is unknown. Our results (Table 1) show that learning from arbitrary pairs with our mechanism achieves an accuracy close to learning only from identical (positive) pairs. This empirically demonstrates that the value of learning cross-class feature subspaces has been largely overlooked by traditional contrastive methods.
> 2) While bringing highly distinct objects together might seem counter-intuitive (and can cause performance drops if executed without control, as noted in Figure 6), this difficult setting highlights the robustness of our gating mechanism.  It is a crucial ability to process large-scale data which is collected from Internet.
> We emphasis our experimental settings in Section 1 (line 294) and Section 4.1.
>
>
>
> > Question (3): If the label names are removed of any semantic meaning at all, would the filtering idea still work?
>
> Yes, the filtering idea would still work. Our original implementation only uses the label indices (e.g., $0, 1, 2, \dots$) to obtain gate vectors in the feature filter. Crucially, these label embeddings are randomly initialized and must learn their semantic meaning entirely from the visual representations propagated through the network. They act as learnable parameters that capture the asymmetric relationships discovered between visual classes, as shown in Figure 4. This highlights the learning is purely driven by visual characteristics from images instead of by textual semantics.
>
> > Question (4): the implementation restricts this subspace as coming from an element-wise product (in other words restricting $A$ to be diagonal)
>
> This is a correct observation regarding the simplicity of our final implementation, but our central idea of learning in subspaces is not limited to a diagonal transformation. We intentionally follow a minimal design to demonstrate our idea of learning from arbitrary pairs.
>
> > Question (5): Is this motivated by sparsity/computational constraints?
>
> No, the motivation is primarily intuitive, rather than being driven by sparsity or computational constraints. No. Our primary motivation comes from the intuitive observation illustrated in Figure 4. We aim to project the global, full-dimensional representations into class-specific subspaces to extract common features that define the similarity between the arbitrary pair. In addition, we did not use any sparse constraint in our work.

---

### Author Response · Authors · 2025-11-25
**Summary of revision**

Dear reviewers,
Thank you for your helpful feedback.
We have revised the manuscript and highlighted every change in blue.
The main update includes:
- Appendix F: added a detailed discussion of our gating mechanism.
- Appendix H: repeated all experiments with additional random seeds.
- Introduction: re-structured the narrative for clarity.
- Section 3.1: sharpened the statement of motivation.
- Some figure captions: re-written to better interpret the displayed results.
- Section 4.1: explicitly emphasized the experimental settings.

---

### Meta-Review · Area_Chair_1GHn · 2026-01-04

**Summary:**

The paper proposes SimLAP, a novel supervised contrastive learning method that leverages class pair-specific subspaces via a lightweight gating module to capture similarities between instances of different classes. The approach is technically sound, clearly presented, and experimentally validated across classification, retrieval, segmentation, and domain-shift benchmarks, with analyses such as embedding visualizations and collapse-prevention studies. Its main strengths lie in the creative use of subspace-based relational representation, a simple and reproducible implementation, and the exploration of rich inter-class relationships beyond standard contrastive setups. However, several weaknesses remain: key design choices, such as the channel-wise gating and averaging of label embeddings, lack clear justification; some claims about superior transfer performance and benefits of learning from arbitrary class pairs are not consistently supported by results; experiments often rely on a single seed and lack variance estimates; comparisons with baselines are inconsistent; and the method is limited to supervised settings without discussion of potential unsupervised extensions. Overall, while the paper introduces an interesting direction for relational representation learning, the empirical evidence is modest, and further analysis is needed to clarify when and why SimLAP provides advantages.

**Reviewer Concerns:**

The authors have provided comprehensive responses to each reviewer’s questions; however, several core issues still require further analysis and experimental validation. For instance, it remains necessary to clarify the main benefits of using the proposed training scheme, to explain the relevance of discussing random Gaussians in the method, and to justify the specific design choices of the feature filter, which is central to the contribution. These questions need to be addressed through systematic experiments with detailed explanations to enhance the interpretability and credibility of the approach.

**Reviewer Scores:**

I believe the reviewers are likely to maintain their original scores. The authors’ responses to each question should be supported and validated through experiments, as this would be far more convincing. Simple textual explanations alone are unlikely to change the reviewers’ judgments.

---

### Decision · Program_Chairs · 2026-01-26

Reject